# FUSE: Ensembling Verifiers with Zero Labeled Data

**Joonhyuk Lee** [* 1]   **Virginia Ma** [* 1]   **Sarah Zhao** [* 1]   **Yash Nair** [1]   **Asher Spector** [1]   **Regev Cohen** [2]
**Emmanuel Candès** [1]

## Abstract

Verification of model outputs is rapidly emerging as a key primitive for both training and real-world deployment of large language models (LLMs). In practice, this often involves using imperfect LLM judges and reward models since ground truth acquisition can be time-consuming and expensive. We introduce Fully Unsupervised Score Ensembling (FUSE), a method for improving verification quality by ensembling verifiers without access to ground truth correctness labels. The key idea behind FUSE is to control conditional dependencies between verifiers in a manner that improves the unsupervised performance of a class of spectral algorithms from the ensembling literature. Despite requiring zero ground truth labels, FUSE typically matches or improves upon semi-supervised alternatives in test-time scaling experiments with diverse sets of generator models, verifiers, and benchmarks. In particular, we validate our method on both conventional academic benchmarks such as GPQA Diamond and on frontier, unsaturated benchmarks such as Humanity's Last Exam and IMO Shortlist questions.

## 1. Introduction

### 1.1. Motivation

Significant recent progress in the performance of large language models (LLMs) has been driven by *test-time scaling*. A key ingredient in many test-time scaling approaches is Best-of-N (BoN) sampling, in which N responses are sampled independently for any given query, and scores from a *verifier* are used to select a single response to return (e.g., Lightman et al., 2023; Cobbe et al., 2021; Nakano et al., 2021; Sun et al., 2024; Rakhsha et al., 2026; Di et al., 2026). Such schemes, for instance, have been credited as a ma-

jor input for the gold-level performance of frontier LLMs at the International Math Olympiad (Luong et al., 2025). The verifiers used in BoN sampling are typically external language models or reward models. While imperfect, such models do not incur the time and monetary costs of acquiring ground-truth labels from human experts. Notably, this practical constraint extends beyond test-time scaling alone— the pipeline of repeated sampling followed by imperfect verification has also embedded itself into numerous techniques for *training* language models, such as reinforcement learning with rubric-based rewards (Gunjal et al., 2025) and synthetic data selection (Liu et al., 2024).

With the imperfect nature of practical verifiers in mind, recent work has proposed *scaling verification* (Zhao et al., 2025; Saad-Falcon et al., 2026), wherein scores for responses are computed by aggregating the outputs of multiple verifiers. In the simplest case, this aggregation involves taking an average or majority-vote across verifiers (Verga et al., 2024; Lifshitz et al., 2025). The performance of such rules depends critically on the quality of the verifiers at hand. As noted by Saad-Falcon et al. (2026), when verifiers exhibit disparate performance, average- and majority-vote-based schemes may perform poorly because they treat scores from unreliable verifiers on par with those given by their stronger counterparts. Further complicating matters is that verifier strength can be query-dependent. These nuances are difficult to adapt to, as by the previously cited constraints of money and time, one often has *zero* ground-truth labels of response correctness for any given query.

### 1.2. FUSE

To address the inherent challenges in constructing an ensemble of verifiers in the absence of ground-truth labels, we develop Fully Unsupervised Score Ensembling (FUSE), a method which aggregates scores from multiple verifiers in a data-adaptive manner to select the most promising LLM-generated response at inference-time. Because FUSE learns how to effectively ensemble verifiers from data, it generally performs better than (sometimes, significantly so) simple unweighted baselines. Despite using zero labeled data, FUSE also performs competitively with prior methods that do assume access to labeled data, and sometimes even outperforms them. Figure 1 summarizes these findings in the

---

[*]Equal contribution  [1]Stanford University [2]Google. Correspondence to: Joonhyuk Lee <joonhyuk@stanford.edu>.

*Proceedings of the 43rd International Conference on Machine Learning*, Seoul, South Korea. PMLR 306, 2026. Copyright 2026 by the author(s).

context of Best-of-N test-time scaling experiments on two datasets considered in Saad-Falcon et al. (2026) and the IMO Shortlist subset of Google's IMO AnswerBench (Luong et al., 2025).

On the methodological side, FUSE builds on the statistical literature on crowd-sourcing and unsupervised ensemble learning, in particular on the works of Parisi et al. (2014); Jaffe et al. (2015). The methods proposed in these papers, which consider unsupervised ensemble learning for binary classification, impose various conditional independence assumptions between classifiers. A key idea behind FUSE is to mitigate the impact of violations of some of these assumptions, which we do not expect to hold in general for LLM verifiers.

Two building blocks, visualized in Figure 2, underlie FUSE:

1. Under an assumption which we refer to as *triplet* conditional independence (TCI), a spectral algorithm in Jaffe et al. (2015) yields consistent estimates of verifier quality. Rather than assuming TCI, we introduce an empirical measure of TCI violations that can be computed without labels. Using this statistic, FUSE learns a transformation of raw verifier scores (Step 1 in Figure 2) for which applying the algorithm of Jaffe et al. (2015) will yield reliable estimates (Figure 2, Step 2).

2. With estimates of verifier quality in hand, we form pseudo-labels that can be used to optimize arbitrary parametrized rules for aggregating verifier scores (e.g. logistic regression) as visualized in Step 3 of Figure 2. Our construction and subsequent use of pseudo-labels allows us to avoid assuming *joint* conditional independence (JCI)—a common assumption used in the crowd-sourcing literature to ensemble classifiers—which is empirically untestable and unlikely to hold in practice.

In summary, FUSE adaptively transforms verifier scores to better satisfy a weaker TCI assumption while bypassing the stronger JCI assumption. By extending Jaffe et al. (2015), we also ensure that all steps in this procedure are compatible with both real and discrete-valued verifiers. Before providing further detail, we first fix notation and give a clear problem. statement.

### 1.3. Problem statement

Given a query $q$, we assume access to $N$ LLM-generated responses $(r_1, \ldots, r_N)$ and a collection of $m$ verifiers $v_1, \ldots, v_m$ that assign correctness scores to each query-response pair. We treat the query as fixed (i.e., non-random), while $(r_1, \ldots, r_N)$ are sampled i.i.d. from $G(q)$, the distribution of the generator model's responses given the prompt $q$. We let $y_i := y(q, r_i) \in \{\pm 1\}$ denote the ground-truth correctness label of the $i$th response for the query:

$y_i = 1$ if and only if $r_i$ is correct for $q$. Finally, letting $v_{i,j} := v_j(q, r_i)$ be the score assigned by the $j$th verifier to response $r_i$, we write $\mathbf{V} := (v_{i,j})_{i,j}$ for the $N \times m$ matrix of verifier scores, and $\mathbf{V}_{i\bullet}$ for the $i$-th row of $\mathbf{V}$.

Our goal is to select, using the score matrix $\mathbf{V}$ alone, a response $r_{i^\star}$ for which $y(q, r_{i^\star}) = 1$. When multiple queries $q_1, \ldots, q_L$ exist, our goal will be to select a correct response for each question. In the latter case, we may pool information from different score matrices $\mathbf{V}_1, \ldots, \mathbf{V}_L$ ('batching') or only use $\mathbf{V}_\ell$ to select responses for $q_\ell$ ('query-conditional'). To minimize notational overhead, we restrict our attention to the single-query case in subsequent sections.

### 1.4. Conflict of Interest Disclosure

R.C. is an employee of Google, which provided partial funding for this research (see Section 5), and is also the developer of the Gemini-3-pro-preview model considered in one of our experimental settings.

## 2. Fully unsupervised score ensembling

### 2.1. MoM estimation of query-specific verifier qualities

The first step in FUSE is to estimate each verifier's accuracy conditioned on the query $q$ using $\mathbf{V}$. When verifiers satisfy strong independence assumptions, we can do so by adapting a method-of-moments (MoM) approach developed by Jaffe et al. (2015) in the context of binary classification. In this section, we provide necessary background for FUSE by explaining this approach, and empirically justify the need to look beyond it. For ease of exposition, we temporarily assume, as in Jaffe et al. (2015), that verifiers output $\{\pm 1\}$-valued predictions. We later relax this assumption, allowing FUSE to work with arbitrary verifiers.

Because all outputs are binary, the quality of any given verifier $v_j$'s predictions on a response $r$ for a query $q$ is determined by its sensitivity $\psi_j$ and specificity $\eta_j$:

$$\psi_j := \mathbb{P}_{r \sim G(q)}(v_j(q, r) = 1 \mid y(q, r) = 1)$$
$$\eta_j := \mathbb{P}_{r \sim G(q)}(v_j(q, r) = -1 \mid y(q, r) = -1)$$

where, as a reminder, $y(q, r)$ is the ground-truth correctness label of the response for the query. Jaffe et al. (2015) propose using the score matrix $\mathbf{V}$ to estimate the verifier sensitivities $\boldsymbol{\psi} := (\psi_j)_{j=1}^m$ and specificities $\boldsymbol{\eta} := (\eta_j)_{j=1}^m$ under the following two assumptions:

**Assumption 2.1** (Majority of classifiers are better than random). Let $\pi_j := \frac{\psi_j + \eta_j}{2}$ denote the *balanced accuracy* of the $j$th verifier and $\boldsymbol{\pi} := (\pi_1, \ldots, \pi_m)$ denote the vector of balanced accuracies for query $q$. Then, more than $\frac{m}{2}$ of the values $\{\pi_j\}_{j=1}^m$ are larger than $1/2$.

**Assumption 2.2** (Triplet conditional independence). Letting $r \sim G(q)$, each triplet of verifiers produces condition-

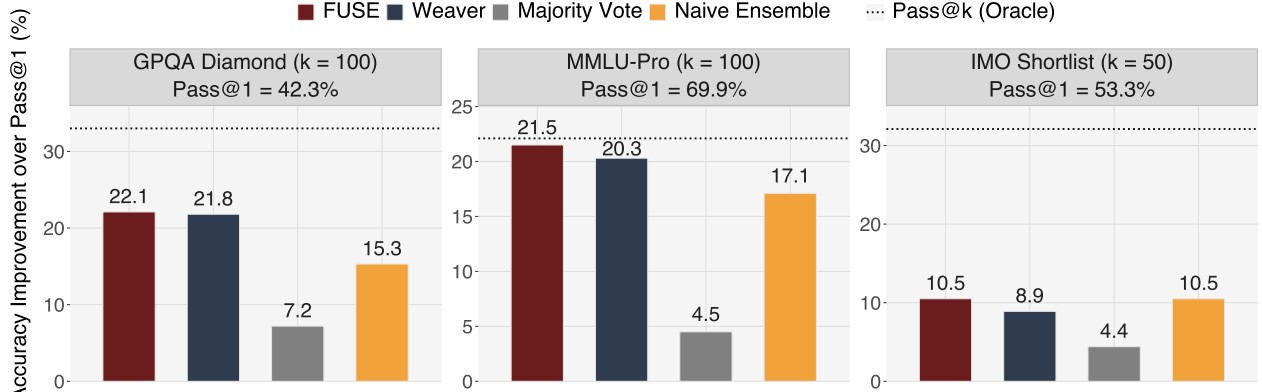

*Figure 1.* BoN accuracy of our method versus that of a leading semi-supervised alternative (WEAVER, by Saad-Falcon et al. (2026)) and unsupervised baselines of naive ensemble and majority vote. All bars are re-scaled to depict improvement over Pass@1, which is the accuracy of a random selection rule. The black dotted Pass@k line denotes the maximum possible accuracy improvement for any selection method. Despite being unsupervised, FUSE is competitive with WEAVER and decisively beats naive ensemble and majority vote in all settings except the IMO Shortlist, where characteristics of our verifier set imply that a naive ensemble is effectively an oracle ensemble (see Section 3.3 for details).

ally independent scores given the ground-truth label $y(q, r)$ (see Appendix A.1 for a precise statement).

Conditions like Assumption 2.1 are common in the crowd-sourcing and ensemble learning literature (c.f., e.g., Dawid & Skene, 1979; Kleindessner & Awasthi, 2018; Shaham et al., 2016; Didwania et al., 2022; Ahsen et al., 2019) and ensure that the problem is identified in the absence of ground-truth labels. In practice, we use certain heuristics—described in Appendix D—to drop verifiers that do not appear to be better than random so as to better ensure Assumption 2.1 is satisfied. Conditional independence assumptions like Assumption 2.2 are also common in these literatures (Dawid & Skene, 1979; Parisi et al., 2014; Tenzer et al., 2022; Jaffe et al., 2015), and also ensure identification. Under these two assumptions, Jaffe et al. (2015) show that verifier sensitivities and specificities can be extracted from a rank-one structure that arises in certain covariance tensors, as summarized by the following:

**Theorem 2.3** (Jaffe et al. (2015, Lemmas 1, 2, and 4)). *Let $\boldsymbol{\mu}$ denote the vector of mean values of verifier predictions for $q$ and $\boldsymbol{\Sigma}$ and $\boldsymbol{T}$ denote the second and third order marginal covariance tensors between verifier outputs for query $q$; i.e., $\boldsymbol{\mu} \in \mathbb{R}^m, \boldsymbol{\Sigma} \in \mathbb{R}^{m \times m}, \boldsymbol{T} \in \mathbb{R}^{m \times m \times m}$ with entries given by $\mu_{j_1} := \mathbb{E}[v_{j_1}(q, r)]$,*

$$\boldsymbol{\Sigma}_{j_1, j_2} = \mathbb{E}\left[\prod_{\ell=1}^{2}(v_{j_\ell}(q, r) - \mathbb{E}[v_{j_\ell}(q, r)])\right],$$

$$\boldsymbol{T}_{j_1, j_2, j_3} = \mathbb{E}\left[\prod_{\ell=1}^{3}(v_{j_\ell}(q, r) - \mathbb{E}[v_{j_\ell}(q, r)])\right],$$

*for all $j_1, j_2, j_3 \in [m]$. Letting $b := \mathbb{P}(y(q, r) = 1) -$*

$\mathbb{P}(y(q, r) = -1)$ *denote the class imbalance for $q$, under Assumptions 2.1 and 2.2:*

*(i) The off-diagonal entries of $\boldsymbol{\Sigma}$ are equal to those of the outer product matrix $\boldsymbol{u}\boldsymbol{u}^\top$, where*

$$\boldsymbol{u} := \sqrt{1 - b^2}(2\boldsymbol{\pi} - 1). \tag{1}$$

*Consequently, $\boldsymbol{u}$ is identifiable from $\boldsymbol{\Sigma}$ up to a sign, which is uniquely determined by Assumption 2.1.*

*(ii) The off-diagonal entries of $\boldsymbol{T}$ (i.e., those with all three indices distinct) are equal to those of the rank-one third order tensor $\boldsymbol{w} \otimes \boldsymbol{w} \otimes \boldsymbol{w}$ where*

$$\boldsymbol{w} := (-2b(1 - b^2))^{1/3}(2\boldsymbol{\pi} - 1). \tag{2}$$

*(iii) The sensitivities and specificities of verifier outputs for responses to query $q$ can be written as:*

$$\psi = \frac{1}{2}\left(1 + \boldsymbol{\mu} + \boldsymbol{u}\sqrt{\frac{1 - b}{1 + b}}\right),$$
$$\eta = \frac{1}{2}\left(1 - \boldsymbol{\mu} + \boldsymbol{u}\sqrt{\frac{1 - b}{1 + b}}\right). \tag{3}$$

Theorem 2.3 shows how the sensitivities $\psi$ and specificities $\eta$ are uniquely determined by the marginal moment tensors $\boldsymbol{\mu}, \boldsymbol{\Sigma}, \boldsymbol{T}$. In particular, parts (i) and (ii) of Theorem 2.3 show how to recover the vectors $\boldsymbol{u}$ and $\boldsymbol{w}$ in equations (1) and (2) from the second and third order covariance tensors. Because these vectors differ only by a scale factor depending on $b$, this in turn recovers the class imbalance $b$. Finally,

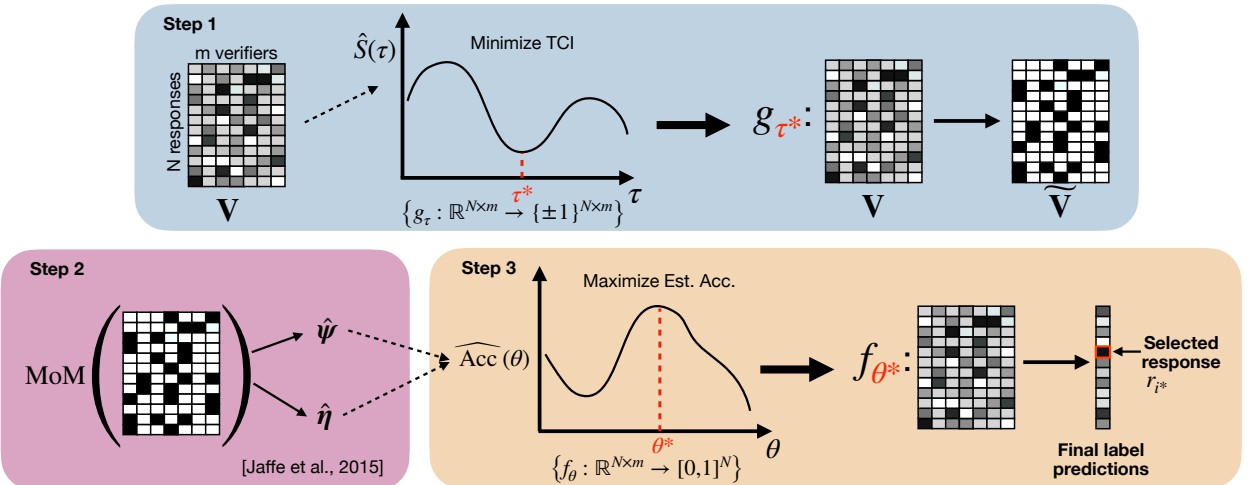

*Figure 2.* Overview of FUSE: given the matrix of verifier scores $\mathbf{V}$ for query $q$, it first finds a transformation $g_{\tau^*}$ that minimizes an empirical measure of TCI violation and transforms scores according to it (**Step 1**). It then uses the moment-based method of Jaffe et al. (2015) to produce estimates of the query-specific sensitivities and specificities $\hat{\boldsymbol{\psi}}, \hat{\boldsymbol{\eta}}$ (**Step 2**). Finally, FUSE uses these estimates to construct an estimated accuracy of any predictor $f_\theta : \mathbb{R}^{N \times m} \to [0, 1]^N$, optimizing this metric across the parametric family $\{f_\theta\}$ ensembles to obtain a final ensemble $f_{\theta^\star}$ and returning the response with highest predicted correctness under $f_{\theta^\star}$ (**Step 3**).

per equation (3), the sensitivities and specificities can be extracted on the basis of $\boldsymbol{u}, b$, and the mean vector $\boldsymbol{\mu}$. Of course, the moment tensors $\boldsymbol{\mu}, \boldsymbol{\Sigma}, \boldsymbol{T}$ are unknown, but their empirical counterparts can be computed from $\mathbf{V}$ and are consistent at $O_p(1/\sqrt{N})$ rates. Consequently, Theorem 2.3 provides a recipe for producing sensitivity and specificity estimates $\hat{\boldsymbol{\psi}}, \hat{\boldsymbol{\eta}}$ with zero ground-truth data.

With such estimates in hand, under the additional assumption that verifiers are jointly conditionally independent, Jaffe et al. (2015) obtain closed-form coefficients for a near-optimal weighted ensemble (see Appendix B for details). Unfortunately, this ensemble does poorly in practice. As depicted in Figure 3, on BoN test-time scaling data from Saad-Falcon et al. (2026), the Jaffe et al. (2015) approach under-performs a simple naive ensemble in 7 out of 10 settings, suggesting that its assumptions are untenable in realistic LLM verification settings. See Figure 4 for conditional correlation plots that confirm this.

### 2.2. Adaptive score transformations for enhanced MoM estimation

To adapt the approach in the previous section to a more general setting in which verifiers may be conditionally dependent and emit real-valued scores, we establish:

**Proposition 2.4.** *Given a set of verifiers, let $\boldsymbol{\Sigma}$ and $\boldsymbol{T}$ denote the second and third order covariance tensors associated with query $q$. Then, under the TCI condition stated in Assumption 2.2,*

$$\sum_{j_3=3}^{m} \mathrm{Var}\left( \left( \frac{\boldsymbol{T}_{j_1,j_2,j_3}}{\boldsymbol{\Sigma}_{j_1,j_2}} \right)_{1 \le j_1 < j_2 < j_3} \right) = 0, \qquad (4)$$

*where* $\mathrm{Var}(\cdot)$ *denotes the sample variance over the indexed collection.*

Proposition 2.4 provides a measure of violations of TCI that does not require ground-truth labels: a large value of the left-hand side of (4) certifies that TCI does not hold. In our setting, we use this statistic as a learning objective to promote TCI across verifiers. Specifically, let $g_j : \mathbb{R} \to \mathbb{R}$ denote a monotone transformation of the $j$th verifier's outputs. Letting $\boldsymbol{g}(\mathbf{V}) := (g_j(v_{i,j}))_{i,j}$ denote the matrix of transformations of the original scores $\mathbf{V}$, we use $\boldsymbol{g}(\mathbf{V})$ to construct empirical estimates of the second- and third-order population covariance tensors between the transformed verifiers $g_1(v_1(\cdot)), \dots, g_m(v_m(\cdot))$. Plugging in these estimates into the left-hand side of (4) yields an empirical approximation of (4)—which we denote by $\hat{\mathcal{S}}(\boldsymbol{g}(\mathbf{V}))$—that provides a (feasible) measure of deviations from TCI among the transformed verifiers.[1]

**Binary transformation** By default, we transform verifiers by binarizing verifier outputs to $\{\pm 1\}$ using a vector $\tau$ of verifier-specific thresholds: $g_{j,\tau_j}(v_j(\cdot)) = \mathrm{sign}(v_j(\cdot) - \tau_j)$. We then find $\tau^\star$ which (approximately) minimizes $\hat{\mathcal{S}}(\tau) := \hat{\mathcal{S}}(\boldsymbol{g}_\tau(\mathbf{V}))$ via coordinate descent and apply the MoM procedure from the previous section on the matrix $\boldsymbol{g}_\tau(\mathbf{V})$ to estimate the sensitivities and specificities of the $\tau^\star$-thresholded verifiers $g_{1,\tau_1^\star}(v_1(\cdot)), \dots, g_{m,\tau_m^\star}(v_m(\cdot))$.[2] Additional examples of

---

[1]In practice, we clip the elements in the denominator of (4) to a small positive number to promote stability.

[2]For faster optimization, gradient descent can be applied by approximating the sign function with a sigmoid.

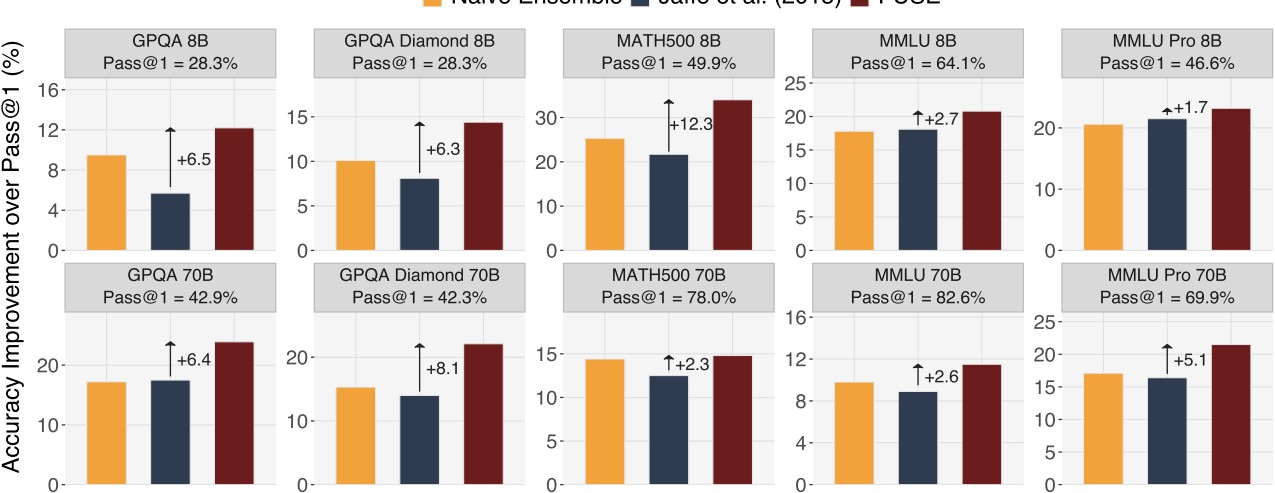

*Figure 3.* Accuracy of Jaffe et al. (2015) versus a naive ensemble and FUSE for response selection on data from Saad-Falcon et al. (2026), in which generator models are Llama 3.3 8B Instruct and Llama 3.3 70B Instruct. All bars are re-scaled to indicate improvement over Pass@1. The black arrow and accompanying number indicates the accuracy gain of FUSE over Jaffe et al. (2015).

transformations are given in Appendix D.

We write $\tilde{\mathbf{V}}$ to denote the transformed matrix of verifier scores (i.e., $\tilde{\mathbf{V}} = \boldsymbol{g}_{\tau^\star}(\mathbf{V})$). Correspondingly, $\tilde{v}_j$ denotes the transformed $j$th verifier (i.e., $\tilde{v}_j(q, r) = g_{j,\tau_j^\star}(v_j(q_{,r}))$). Applying the MoM procedure of Section 2.1 to the matrix $\tilde{\mathbf{V}}$ then yields estimates $\hat{\boldsymbol{\psi}}, \hat{\boldsymbol{\eta}}$ of the $q$-conditional sensitivities and specificities of the transformed verifiers $\tilde{v}_1, \ldots, \tilde{v}_m$.

### 2.3. Ensemble construction

The final step of FUSE is to use the estimated verifier sensitivities and specificities to construct an ensemble. We will again depart from Jaffe et al. (2015) who impose the stronger JCI assumption to construct an ensemble. Rather, we will use the estimates $\hat{\boldsymbol{\psi}}, \hat{\boldsymbol{\eta}}$ to measure the quality of any given ensemble and subsequently optimize this estimated quality measure in a manner that is not tied to JCI. To illustrate the idea, temporarily suppose that the posterior label probability

$$p^\star(r) := \mathbb{P}(y(q, r) = 1 \mid \tilde{v}_1(q, r), \ldots, \tilde{v}_m(q, r)) \quad (5)$$

for query $q$ were known. Now, consider any family of predictors $f_\theta$ indexed by parameters $\theta$. Concretely, in our experiments we take $f_\theta$ to be the class of logistic regression classifiers operating on the design matrix $\mathbf{V}$. Which choice of parameters $\theta$ leads to the most accurate predictions of the ground-truth labels? We propose to measure the quality of any given $\theta$ using the following objective

$$\sum_{i=1}^{N} (2p^\star(r_i) - 1)\hat{f}_\theta(\mathbf{V}_{i\bullet}), \quad (6)$$

where $\hat{f}_\theta(\mathbf{V}_{i\bullet})$ is the $\{\pm 1\}$-valued prediction of $f_\theta$ given the (original) verifier outputs for the $i$th response (i.e., the $i$th row of $\mathbf{V}$). Intuitively, parameter values which achieve larger values of this objective correspond to better ensembles. Of course, the objective in (6) is infeasible to compute since the probabilities $p^\star(r_i)$ are unknown. Therefore FUSE substitutes these oracle probabilities with estimates $\hat{p}(r_i)$ derived from the estimated sensitivities/specificities $\hat{\boldsymbol{\psi}}, \hat{\boldsymbol{\eta}}$—we will soon describe precisely how the estimates $\hat{p}(r_i)$ are computed from $\hat{\boldsymbol{\psi}}, \hat{\boldsymbol{\eta}}$. In particular, the final ensemble that FUSE constructs is given by $f_{\theta^\star}$, where

$$\theta^\star := \arg\max_\theta \sum_{i=1}^{N} (2\hat{p}(r_i) - 1)\hat{f}_\theta(\mathbf{V}_{i\bullet}) =: \widehat{\mathrm{Acc}}(\theta). \quad (7)$$

Once $\theta^\star$ has been computed, FUSE selects the final response to generate as the one with the greatest likelihood of being correct as measured by $f_{\theta^\star}$; i.e., $r_{i^\star}$ where $i^\star := \arg\max_{i=1}^{N} f_{\theta^\star}(\mathbf{V}_{i\bullet})$.

**Posterior probability estimation** To construct estimates $\hat{p}(r_i)$ of the posterior probabilities, we first estimate the posterior probabilities of the correctness label given any three verifiers. A direct calculation shows that for any three verifiers $\tilde{v}_{j_1}, \tilde{v}_{j_2}, \tilde{v}_{j_3}$, the posterior label probability

$$\mathbb{P}(y(q, r) = 1 \mid \tilde{v}_{j_1}(q, r), \tilde{v}_{j_2}(q, r), \tilde{v}_{j_3}(q, r)) \quad (8)$$

can be expressed, under TCI, in terms of the sensitivities $\psi_{j_1}, \psi_{j_2}, \psi_{j_3}$, specificities $\eta_{j_1}, \eta_{j_2}, \eta_{j_3}$, and class imbalance $b$. The exact form of this relationship is given in Proposition C.1 in Appendix C. Plugging in estimates of label imbalance $\hat{b}$ as well as sensitivity and specificity estimates

$\hat{\boldsymbol{\psi}}, \hat{\boldsymbol{\eta}}$ then results in an estimate of this posterior label probability: we refer to this estimate as $\hat{p}_{j_1,j_2,j_3}(r)$. Unfortunately, $\hat{p}_{j_1,j_2,j_3}(r)$ may in general be a poor estimate of the *full* posterior label probability $p^\star(r)$. Our default remedy (see Appendix E for alternatives) is to average posterior estimates across triplets. That is, we estimate $p^\star(r)$ as:

$$\hat{p}(r) = \frac{1}{\binom{m}{3}} \sum_{1 \le j_1 < j_2 < j_3 \le m} \hat{p}_{j_1,j_2,j_3}(r). \qquad (9)$$

See Figure 2 for a high-level visualization of FUSE, and Algorithm 1 for pseudocode. Finally, for an extension to the 'batched' case in which scores for multiple queries are jointly used to select responses, see Appendix D.

---

**Algorithm 1** Fully Unsupervised Score Ensembling

---

1: **Input:** Score matrix $\mathbf{V} \in \mathbb{R}^{N \times m}$ of verifier scores for responses $(r_i)_{i=1}^N$ to query $q$; user-defined threshold family $\{g_\tau : \mathbb{R}^{N \times m} \to \{\pm 1\}^{N \times m}\}$; user-specified ensemble family $\{f_\theta : \mathbb{R}^{N \times m} \to [0,1]^N\}$.
2: Compute $\tau^\star \in \arg\min_{\tau \in \mathcal{T}} \hat{\mathcal{S}}(g_\tau(\mathbf{V}))$.
3: Set $\tilde{\mathbf{V}} \leftarrow g_{\tau^\star}(\mathbf{V}) \in \{\pm 1\}^{N \times m}$.
4: Apply Theorem 2.3 to $\tilde{\mathbf{V}}$ to obtain $(\hat{\boldsymbol{\psi}}, \hat{\boldsymbol{\eta}}, \hat{b})$.
5: Define the estimated ensemble accuracy $\widehat{\text{Acc}}(\theta)$ in (7) using (9).
6: Compute $\theta^\star \in \arg\max_{\theta \in \Theta} \widehat{\text{Acc}}(\theta)$.
7: **Return** $r_{i^\star}$ where $i^\star \leftarrow \arg\max_{i \in [N]} f_{\theta^\star}(\mathbf{V}_{i\bullet})$.

---

## 3. Results

In this section, we compare the performance of FUSE to that of semi-supervised and unsupervised baselines in the context of BoN test-time scaling. Our basic setup, detailed further in Appendix E, is as follows:

**Baselines** We consider three semi-supervised baselines, which are each given access to ground truth labels for 5% of queries. These are logistic regression, naive Bayes, and WEAVER (Saad-Falcon et al., 2026), which uses ideas from the weak supervision literature to estimate ensemble weights. Unsupervised baselines include majority vote, which selects the most common answer among the repeated generations, and naive ensemble, which selects the response with the highest average verifier score. Finally, we compute several oracle baselines such as Pass@$k$ and the best verifier by ground-truth accuracy. Detailed baseline descriptions are in Appendix E.1. In Appendix E.7, we mathematically justify omitting repeated sampling from a single verifier, which is an obvious but non-competitive baseline. Finally, Appendix E.8 compares FUSE to additional unsupervised baselines such as Dawid & Skene (1979).

**Models and datasets** We consider three data and model sources.

- **Data from Saad-Falcon et al. (2026)** We run our method on the exact data used in Saad-Falcon et al. (2026), consisting of 100 generations per question by Llama 3.1 8B Instruct and Llama 3.3 70B Instruct on GPQA, GPQA Diamond, MATH500, MMLU, and MMLU Pro, with verifications by up to 33 open-source reward models and language models.

- **Humanity's Last Exam** A 649-question subsample of Humanity's Last Exam (Phan et al., 2025). See Appendix E.4 for details on construction. We sample 50 generations per question from Gemini 3 Pro Preview, and verifications from seven closed-source and open-source models (see Appendix E.4).

- **IMO Shortlist** A 123-question subset of IMO AnswerBench (Luong et al., 2025) comprised of modified IMO Shortlist questions. We sample 50 generations per question from the open-source model Qwen3-30B-A3B-Thinking-2507, and verifications from 9 open-source language models.

An attractive feature is that these settings exhibit vast differences in task difficulty and in the strength of generator and verifier models. Consequently, our method's strong overall performance provides evidence for usefulness across problem ranges. In an additional suite of ablations, we verify that the ability of our method to operate on a prompt-conditional basis improves performance versus baselines in 'mixed' settings with task heterogeneity.

### 3.1. 70B and 8B experiments

Table 1 summarizes the Best-of-100 performance of FUSE and baselines on data from Saad-Falcon et al. (2026).

**FUSE is competitive with semi-supervised baselines** In the 70B setting, we see that FUSE achieves essentially exact parity with WEAVER, with minor deviations in performance across benchmarks (for instance, 64.4% on GPQA Diamond for FUSE vs 64.1% for WEAVER). Overall, across 70B and 8B settings, FUSE wins 27 out of 40 comparisons against supervised baselines, and always outperforms the natural unsupervised baseline of naive ensemble and majority vote. We attribute the ability of FUSE to reach parity with baselines involving supervision to distinct failure modes of the latter. For instance, our conditional correlation plots in Appendix E.2 show that verifiers are typically strongly conditionally correlated, so baselines like WEAVER and Naive Bayes that assume conditional independence likely suffer from model misspecification. For the oracle best verifier (OBV), relying solely on the single most accurate verifier may discard potentially useful signals from the remaining ones. Finally, for logistic regression, the small training sample size (5% of each benchmark) may induce high variance.

*Table 1.* Best-of-100 accuracies for all baselines on data from Saad-Falcon et al. (2026), wherein generator models are Llama 3.1 8B Instruct and Llama 3.3 70B Instruct. Supervised methods are given access to ground-truth labels for all 100 responses to 5% of questions (e.g. 2500 labels on MATH500). The OBV (Oracle Best Verifier) column corresponds to the selection accuracy of the verifier with highest balanced accuracy, which may vary from benchmark to benchmark.

| Benchmark (70B) | Unsupervised | | | | Supervised | | | | Oracle |
| | Pass@1 | Majority Vote | Naive Ensemble | FUSE | OBV | Weaver | Logistic | Naive Bayes | Pass@100 |
| --- | --- | --- | --- | --- | --- | --- | --- | --- | --- |
| GPQA | 42.9 | 47.4 | 60.1 | **66.8** | 59.1 | 66.4 | **69.3** | 60.5 | 81.0 |
| GPQA Diamond | 42.3 | 49.5 | 57.6 | **64.4** | 50.8 | **64.1** | 63.3 | 57.1 | 75.3 |
| MATH500 | 78.0 | 82.4 | 92.4 | **92.8** | 87.2 | 93.4 | **96.2** | 94.4 | 98.6 |
| MMLU | 82.6 | 84.1 | 92.4 | **94.1** | 88.4 | **94.9** | 93.1 | 93.7 | 96.0 |
| MMLU Pro | 69.9 | 74.4 | 87.0 | **91.4** | 85.6 | 90.2 | **91.8** | 91.4 | 92.0 |

| Benchmark (8B) | Unsupervised | | | | Supervised | | | | Oracle |
| | Pass@1 | Majority Vote | Naive Ensemble | FUSE | OBV | Weaver | Logistic | Naive Bayes | Pass@100 |
| --- | --- | --- | --- | --- | --- | --- | --- | --- | --- |
| GPQA | 28.3 | 30.5 | 37.8 | **40.5** | 41.9 | **47.1** | 35.9 | 37.6 | 95.2 |
| GPQA Diamond | 28.3 | 32.3 | 38.4 | **42.7** | 40.6 | **46.5** | 34.3 | 37.5 | 95.0 |
| MATH500 | 49.9 | 69.6 | 75.2 | **83.9** | 82.8 | 74.8 | **88.0** | 80.8 | 99.2 |
| MMLU | 64.1 | 72.7 | 81.9 | **84.9** | 83.6 | **85.7** | 80.4 | 77.8 | 98.5 |
| MMLU Pro | 46.6 | 56.4 | 67.2 | **69.8** | 66.5 | **67.2** | 65.2 | 66.0 | 96.8 |

**Diverse verification is necessary for strong performance** Across all benchmarks, the verification-free baseline of majority vote is uniformly non-competitive, with gaps in performance versus FUSE that range from $10.0\%$ on 8B GPQA to $17.0\%$ on MMLU Pro. Further, FUSE outperforms the oracle 'best verifier' in all but one setting, suggesting that using diverse verifiers produces meaningful benefits.

*Table 2.* Performance of methods in a mixed-data setting with 100 questions each from GPQA, GPQA Diamond, MATH500, MMLU, and MMLU Pro. In the mixed labels setting, 5 questions from each benchmark are held out as a labeled train set. In the GPQA-only setting, 25 questions from GPQA are held out.

| Setting | Method | 8B acc. | 70B acc. |
| --- | --- | --- | --- |
| Mixed Labels | Pass@1 | 40.2% | 64.0% |
| | Pass@100 | 97.1% | 87.6% |
| | Majority Vote | 52.2% | 67.1% |
| | Naive Ensemble | 63.0% | 79.4% |
| | WEAVER | 60.1% | 78.3% |
| | FUSE | **64.4%** | **81.8%** |
| GPQA-Only Labels | Pass@1 | 41.1% | 65.3% |
| | Pass@100 | 97.1% | 88.8% |
| | Majority Vote | 53.5% | 68.0% |
| | Naive Ensemble | 64.2% | 80.8% |
| | WEAVER | 58.4% | 79.6% |
| | FUSE | **65.2%** | 82.4% |

**Prompt-level conditioning bolsters performance in settings with high heterogeneity** We conduct a mixed-data ablation to simulate the effects of high task heterogeneity and potential distribution shift; see Table 2. Our dataset con-

sists of the first 100 questions from each of GPQA, GPQA Diamond, MATH500, MMLU, and MMLU Pro. In the 'Mixed Labels' setting, semi-supervised methods are given labels from the first 5 questions in each benchmark (5% total). In the 'GPQA-Only' setting, labels are derived exclusively from GPQA. We see that FUSE outperforms all baselines in all settings, and that in the 8B setting, transitioning from 'Mixed Labels' to 'GPQA-Only' widens the gap between FUSE and WEAVER from 4.3% to 6.8%.

### 3.2. Humanity's Last Exam

**FUSE performs well in extremely hard settings** The experimental setup in Saad-Falcon et al. (2026) consists of benchmarks which, while standard, are not commensurate with the reasoning quality of recent language models. We assess FUSE and baselines on a curated 649-question subset of Humanity's Last Exam (Phan et al., 2025), which is presently unsaturated by frontier closed-source models such as Gemini 3 Pro and GPT 5.2 Pro. In constructing this subset, we exclude questions with zero correct responses so as to make comparisons between selection methods meaningful.

In Table 3, only the oracle best verifier, semi-supervised logistic regression, and FUSE (using 7 verifiers) outperform the Pass@1 baseline. Notably, naive ensemble is *worse* than simply selecting a random response, validating the extreme difficulty of this setting.

*Table 3.* Best-of-50 performance of methods on Humanity's Last Exam.

| Method | Accuracy |
|---|---|
| Pass@1 | 52.1% |
| Naive Ensemble | 51.4% |
| Oracle Best Verifier (GPT-5.2 High) | 53.5% |
| Naive Bayes | 52.0% |
| Logistic Regression | 53.4% |
| WEAVER | 51.2% |
| **FUSE** | **54.3%** |

*Table 4.* Best-of-50 performance of methods on IMO Shortlist

| Method | Accuracy |
|---|---|
| Pass@1 | 53.3% |
| Majority Vote | 57.7% |
| Naive Ensemble | **63.8%** |
| Oracle Best Verifier (gpt-oss-20b) | 59.7% |
| Naive Bayes | 59.1% |
| Logistic Regression | 60.2% |
| WEAVER | 62.1% |
| **FUSE** | **63.8%** |

### 3.3. IMO Shortlist

In our final main setting, we consider the IMO Shortlist subset of IMO AnswerBench (Luong et al., 2025), which consists of past IMO Shortlist questions that are expert-modified to prevent memorization from training data. This setting, in contrast to the previous ones, features substantial homogeneity in verifier strength and near-conditional independence of verifiers (see Appendix E.5). As such, the naive ensemble baseline is effectively an oracle baseline. We see in Table 4 that FUSE is the only method to match a naive ensemble, while other data-dependent methods fall behind. This setting, besides having highly similar verifiers, also has substantially fewer verifiers than the Saad-Falcon et al. (2026) experiments (9 vs 33). Therefore, neither a high verifier count nor extreme verifier heterogeneity are required for our method to perform well.

## 4. Related work

### 4.1. Verification for test-time scaling

A large number of works have studied variants of BoN for improved test-time scaling when using a single verifier model (e.g., Cobbe et al., 2021; Nakano et al., 2021; Ichihara et al., 2025; Jinnai et al., 2024; Rakhsha et al., 2026). Like these works, FUSE aims to achieve response accuracy as close as possible to the Pass@$k$ rate (Chen, 2021), but differs in that it uses *multiple* verifier models to boost

accuracy.

Prior work on leveraging multiple verifiers to improve verification quality includes Verga et al. (2024); Lifshitz et al. (2025); Saad-Falcon et al. (2026), who propose various unweighted and semi-supervised rules for score aggregation. Such rules are either too simple to account for differing abilities and statistical dependencies between verifiers or rely on holdout sets of labeled data. In contrast, FUSE constructs effective and data-adaptive ensembles with zero access to ground truth labels. This has some conceptual (albeit non-technical) similarity to Qiu et al. (2026), who show that game-theoretic ideas can be used to enhance peer-prediction in the absence of ground-truth labels. Lastly, we remark that outside of the test-time scaling setting, Coste et al. (2024); Eisenstein et al. (2024) study how ensembling verifiers can reduce reward-hacking in reinforcement learning.

### 4.2. Unsupervised ensembling

As detailed in Section 2, the core methodological ideas underlying FUSE stem from the literature on ensembling ML models with no access to labeled data (Parisi et al., 2014; Jaffe et al., 2015; 2016; Tenzer et al., 2022). More generally, weak supervision methods (Ratner et al., 2017) attempt to learn a joint distribution over true labels and user-specified noisy labeling functions without observing the former. These works all require conditional independence assumptions which empirically fail to hold in the LLM verification setting, whereas we adaptively manage such violations so that a Jaffe et al. (2015)-inspired estimation procedure can be applied.

## 5. Discussion

We introduced FUSE, a method for unsupervised ensembling of verifiers in settings where one has access to repeated samples from a generator model. Our experiments involve settings which range from conventional (e.g. MMLU) to frontier (e.g. Humanity's Last Exam) in difficulty. Regardless of setting, FUSE competes well with baselines that require ground-truth labels. An attractive feature of our method, which also boosts performance when tasks are diverse, is that FUSE supports both query-conditional and batched modes of operation.

The primary limitation of our method is its computational and logistical efficiency. As presented, our method is cumbersome in that it demands sampling from distinct verifier models. In Saad-Falcon et al. (2026), distillation of ensemble predictions into a small model is proposed as way to reap the benefits of diverse verification in a FLOP-efficient manner. A similar solution could be implemented in our setting, but any fixed model would be susceptible to distribution shift, whereas the question-conditional version of

FUSE is immune by design.

We expect our ideas to have applicability beyond the test-time scaling setting, and record some promising directions below:

- **Unsupervised model ranking** Robust scoring of responses without ground-truth labels may enable unsupervised model ranking and exciting new forms of benchmarks. For instance, Nie et al. (2025) propose benchmarking language models through LLM-based scoring of attempts at unsolved questions. The credibility of this scheme is unavoidably tied to the strength of available scoring protocols, which FUSE and future refinements would boost.

- **Benchmark and ground-truth auditing** Many benchmarks have label errors which require human experts to identify, for which FUSE could provide diagnostics by isolating inconsistencies between answer key correctness and estimated correctness. Similarly, FUSE could be used to rank *sources* of ground-truth such as expert human labelers on challenging tasks.

- **Data filtering** Unsupervised, high-quality scoring of model generations is of potential relevance in any task where data filtering is necessary, such as selection of synthetic training data and reinforcement learning with rubric-based rewards.

For many tasks, model capability and the cost of useful ground-truth labels will inevitably rise in tandem. The most ambitious conceptual response to this marriage of cost and utility is dispensing with supervision altogether. We do not expect this to be possible in all circumstances and settings, but view FUSE as providing initial validation that the full potential of unsupervised methods lies beyond the present frontier.

## Acknowledgments

We thank Andrew Ilyas, Sarah Cen, and Zitong Yang for helpful discussions. This work was partially conducted using the Stanford Marlowe GPU cluster (Kapfer et al., 2025). Y.N. and A.S. were both supported by a National Science Foundation Graduate Research Fellowship. A.S. was also supported by the Citadel GQS PhD fund. V.M. was supported by the Stanford Data Science Scholarship. E.J.C. was supported by the Office of Naval Research (grant N00014-24-1-2305) and the National Institutes of Health (grant 1R01AG08950901A1). We gratefully acknowledge the support of Google and Google Cloud.

## Impact Statement

This paper presents work whose goal is to advance the field of Machine Learning. There are many potential societal consequences of our work, none which we feel must be specifically highlighted here.

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

## A. Further details on MoM estimation of sensitivities and specificities

In this section, we elaborate on further details of the MoM estimation procedure of Jaffe et al. (2015). Section A.1 formalizes the TCI condition in Assumption 2.2 and Section A.3 explains how the procedure and surrounding results can be extended to real-valued scores.

### A.1. Triplet conditional independence

Formally, the TCI condition stated in Assumption 2.2 is that, for each triplet of distinct indices $j_1, j_2, j_3 \in [m]$ and every $a_{j_1}, a_{j_2}, a_{j_3}, y \in \{\pm 1\}$, we have

$$
\begin{aligned}
\mathbb{P}(v_{j_1}(q,r) = a_{j_1}, v_{j_2}(q_k, r) = a_{j_2}, v_{j_3}(q_k, r) = a_{j_3} \mid y(q,r) = y) = {} & \mathbb{P}(v_{j_1}(q,r) = a_{j_1} \mid y(q,r) = y) \\
& \times \mathbb{P}(v_{j_2}(q,r) = a_{j_2} \mid y(q,r) = y) \quad (10) \\
& \times \mathbb{P}(v_{j_3}(q,r) = a_{j_3} \mid y(q,r) = y).
\end{aligned}
$$

### A.2. Estimation of class imbalance

For completeness and clarity, we discuss the procedure presented in Jaffe et al. (2015) to estimate class imbalance $\mathbf{b}$. In particular, note that since $\mathbf{u} = \sqrt{1 - b^2}(2\pi - 1)$ (Eq (1)) and $\mathbf{w} = (-2b(1 - b^2))^{1/3}(2\pi - 1)$, $T = \alpha(b)\,\mathbf{u} \otimes \mathbf{u}$ on the off-diagonal, where $\alpha(b) := -2b/(1 - b^2)^{1/2}$. Using this relation, we first obtain $\hat{\mathbf{u}}$ from $\hat{\Sigma}$, then obtain an estimate $\hat{\alpha} := \arg\min_\alpha \sum_{j_1 < j_2 < j_3}(\hat{T}_{j_1 j_2 j_3} - \alpha \hat{u}_{j_1} \hat{u}_{j_2} \hat{u}_{j_3})^2$, and finally solve for $b$ via $\hat{b} = -\dfrac{\hat{\alpha}}{\sqrt{4 + \hat{\alpha}^2}}$.

### A.3. MoM estimates for real-valued score matrices

As discussed in Section 2.1, the MoM estimator of Jaffe et al. (2015) can be extended to real-valued scores. We operate under the assumption that all verifier scores lie in the interval $[-1, 1]$.[3] Furthermore, we extend the definition of sensitivity and specificity in this setting to be:

$$
\psi_j := \mathbb{E}\left[\frac{1 + v_j(q,r)}{2} \mid y(q,r) = 1\right], \quad \eta_j := \mathbb{E}\left[\frac{1 - v_j(q,r)}{2} \mid y(q,r) = -1\right].
$$

As before, the balanced accuracy is given by $\pi_j := \frac{\psi_j + \eta_j}{2}$. Notice that in the original $\{\pm 1\}$-valued case, these definitions coincide with those presented in the main manuscript. Finally, we require the same TCI condition as in Assumption 2.2: the (real-valued) scores output by any three distinct verifiers are conditionally independent given the true label. We claim that, under these definitions and conditions, Theorem 2.3—which restates Jaffe et al. (2015)'s identities relating $\psi$ and $\eta$ to the first three moment/covariance tensors in the binary classification setting—continues to hold in this real-valued setup. The reason is that the proofs in Parisi et al. (2014); Jaffe et al. (2015) use only the fact that $\psi_j$ and $\eta_j$ are equal to expectations of (affine transformations of) verifier outputs. This relationship is entirely unchanged in our setting, meaning that their proofs continue to go through in our real-valued setting without modification. To illustrate the idea, we provide a proof of the extension of part (i) of Theorem 2.3.

*Proof of real-valued extension of Theorem 2.3 (i).* It suffices to verify that, for any $j_1 \neq j_2$,

$$
\mathbb{E}[(v_{j_1}(q,r) v_{j_2}(q,r)] - \mathbb{E}[v_{j_1}(q,r)]\mathbb{E}[v_{j_2}(q,r)] = (1 - b^2)(2\pi_{j_1} - 1)(2\pi_{j_2} - 1). \tag{11}
$$

Under TCI, the left-hand side equals

$$
\begin{aligned}
& \frac{1 + b}{2} \cdot (2\psi_{j_1} - 1)(2\psi_{j_2} - 1) + \frac{1 - b}{2} \cdot (2\eta_{j_1} - 1)(2\eta_{j_2} - 1) \\
& \quad - \left[\frac{1 + b}{2} \cdot (2\psi_{j_1} - 1) - \frac{1 - b}{2} \cdot (2\eta_{j_1} - 1)\right]\left[\frac{1 + b}{2} \cdot (2\psi_{j_2} - 1) - \frac{1 - b}{2} \cdot (2\eta_{j_2} - 1)\right]
\end{aligned}
$$

A direct calculation shows that this is the same as the right-hand side of (11). □

---

[3]Accordingly, we re-scale verifier scores to lie in $[-1, 1]$ before applying FUSE.

Finally, we note that Proposition 2.4 continues to hold in our real-valued setting, as it is directly implied by (the real-valued extension of) Theorem 2.3.

## B. Ensembling under joint conditional independence (Jaffe et al., 2015)

Suppose that the true sensitivities and specificities $\psi, \eta$ are known. Then—again, translating Jaffe et al. (2015)'s results and setup to our repeated verification setting—given a response $r$ to query $q$, Jaffe et al. (2015) propose a procedure to estimate the unknown label $y(q, r)$. In particular, under the JCI assumption that the verifier scores $v_j(q, r)$ are *jointly* conditionally independent given the true label $y(q, r)$, they show that the maximum likelihood estimate (MLE) of $y(q, r)$ is given by

$$\hat{y} := \text{sign}\left( \sum_{j=1}^{m} v_j(q, r) \log\left( \frac{\psi_j(1 - \psi_j)}{\eta_j(1 - \eta_j)} \right) + \log\left( \frac{\psi_j(1 - \psi_j)}{\eta_j(1 - \eta_j)} \right) \right), \tag{12}$$

As $\psi$ and $\eta$ are unknown in practice, Jaffe et al. (2015) propose to simply plug in the estimates $\hat{\psi}, \hat{\eta}$ into (12) to obtain an approximation of the MLE.

## C. Estimation of posterior probabilities

In this section, we show how to estimate the posterior label probabilities given any three verifier predictions.

**Proposition C.1.** *For any three indices $j_1, j_2, j_3 \in [m]$, we have that*

$$\mathbb{P}(y(q, r_i) = y \mid v_{i,j_1}, v_{i,j_2}, v_{i,j_3}) \propto (1 + by) \prod_{\ell=1}^{3} [1 - yv_{j_\ell} + v_{j_\ell}((1+y)\psi_{j_\ell} - (1-y)\eta_{j_\ell})] \tag{13}$$

*under the TCI condition (Assumption 2.2). Consequently, plugging in the estimates of class imbalance, sensitivities, and specificities above leads to a consistent estimate of the posterior probability $\mathbb{P}(y(q, r_i) = y \mid v_{i,j_1}, v_{i,j_2}, v_{i,j_3})$.*

*Proof.* The proof follows by a direct calculation:

$$\mathbb{P}(y(q, r_i) = 1 \mid v_{i,j_1}, v_{i,j_2}, v_{i,j_3}) \propto \mathbb{P}(v_{i,j_1}, v_{i,j_2}, v_{i,j_3} \mid y(q, r_i) = 1)\mathbb{P}(y(q, r_i) = 1)$$

$$= \mathbb{P}(y(q, r_i) = 1) \prod_{\ell=1}^{3} \mathbb{P}(v_{j_\ell}(q, r_i) \mid y(q, r_i) = 1)$$

$$= \mathbb{P}(y(q, r_i) = 1) \prod_{\ell=1}^{3} \left[ \frac{1 + v_{j_\ell}}{2} \psi_{j_\ell} + \frac{1 - v_{j_\ell}}{2}(1 - \psi_{j_\ell}) \right]$$

$$= \left( \frac{1 + b}{2} \right) \prod_{\ell=1}^{3} \left[ v_{j_\ell}\psi_{j_\ell} + \frac{1 - v_{j_\ell}}{2} \right],$$

where the first equality holds by TCI. Similarly,

$$\mathbb{P}(y(q, r_i) = -1 \mid v_{i,j_1}, v_{i,j_2}, v_{i,j_3}) \propto \left( \frac{1 - b}{2} \right) \prod_{\ell=1}^{3} \left[ \frac{1 + v_{j_\ell}}{2}(1 - \eta_{j_\ell}) + \frac{1 - v_{j_\ell}}{2}\eta_{j_\ell} \right]$$

$$= \left( \frac{1 - b}{2} \right) \prod_{\ell=1}^{3} \left[ \frac{1 + v_{j_\ell}}{2} - v_{j_\ell}\eta_{j_\ell} \right].$$

In summary, we have that

$$\mathbb{P}(y(q, r_i) = y \mid v_{i,j_1}, v_{i,j_2}, v_{i,j_3}) \propto (1 + by) \prod_{\ell=1}^{3} [1 - yv_{j_\ell} + v_{j_\ell}((1+y)\psi_{j_\ell} - (1-y)\eta_{j_\ell})],$$

as was to be shown. □

# D. Additional details for FUSE

In this section, we record additional implementation details and design considerations.

**Batching** As mentioned in the main text, our method can operate in both query-conditional and batched modes. When batching, we vertically concatenate score matrices $\mathbf{V}_1, \ldots, \mathbf{V}_\ell$ corresponding to queries $q_1, \ldots, q_\ell$ to form a 'tall' score matrix $\mathbf{V}$ and apply FUSE to $\mathbf{V}$ to learn an ensemble. Then, for each query $q$, we return the response with the highest predicted probability of correctness in the corresponding sub-matrix of $\mathbf{V}$.

**Dropping** Various heuristics can be used to drop potentially poor verifiers. By default, we use a balanced-accuracy criterion. After obtaining the transformed matrix $\mathbf{V}$, we apply the method-of-moments estimate in Jaffe et al. (2015) to the entire matrix, and drop verifiers with estimated balanced accuracy less than $\frac{1}{2}$ before proceeding with posterior estimation. While informally motivated, we find this to be robustly performance-enhancing. Notably, even prior works that focus on theoretical guarantees (e.g., Tenzer et al. (2022)) use similar dropping heuristics in real-data settings.

**Posterior aggregation** An alternative approach to aggregating posterior estimates from triplets is to *merge* verifiers prior to transformation. Intuitively, if a 'verifier' is the average of scores from several other verifiers, a triplet including it will incorporate information from more than three verifiers. In the extreme case, one can imagine condensing all verifiers into a single triplet. This approach was inspired by Steinhardt & Liang (2016), who assume in the context of unsupervised risk estimation that a variable set can be partitioned into three 'views'. In practice, we find simple averaging to be superior unless the number of verifiers is large.

**Cross-fitting** In principle, one can form the expected accuracy objective (7) through cross-fitting—splitting the $N$ responses into random folds, and ensuring that posterior estimates and predictors see disjoint data. Because our setting is *transductive*—we are not interested in generalization, and exclusively care about predicting an in-sample label—the traditional statistical intuition that one must avoid label leakage has less force here. In practice, for the range of $N$ considered in this work, we do not find that cross-fitting produces reliable improvements in selection accuracy, and hence choose to omit it by default.

**Predictors** In principle, once pseudo-labels have been acquired, a natural and parameter-free selection rule exists—selecting the response with the highest probability of correctness according to the pseudo-labels. The primary advantage of using pseudo-labels to fit an alternative predictor is therefore not that this makes selection possible, but that the fitting process can (i) act as a form of regularization (ii) encode favorable inductive biases and (iii) allow one to incorporate auxiliary information or covariates that the pseudo-labeling process does not have access to. These reasons, while not fully explored in the present text (e.g., we do not use auxiliary covariates in any of our experiments), may nonetheless enhance the general applicability of FUSE.

Further, given that we default to using logistic regression as the predictor class in our experiments, one may wonder if we should optimize the cross-entropy loss instead of the average accuracy (6). We find that this choice makes little difference; see Table 11.

**Alternative Transformations** Since real-valued or rubric-valued scores may encode richer information than binary ones, it can be desirable to apply real-valued transformations that preserve this granularity.[4] One way of doing so is to use Box-Cox transformations (Box & Cox, 1964).

## D.1. Computational Efficiency

In practice, we expect generation of verifier scores to always be the dominant computational cost in running FUSE. That is, the main cost is not FUSE itself, but the acquisition of the data it operates on. Indeed, the data collection portion of our experiments require either modern GPUs or API spend; see Appendix E and F for details.

Nonetheless, we also found all non-coordinate descent steps in FUSE to be essentially instantaneous on modern, non-specialized hardware (e.g. laptop CPUs). Much of this is attributable to the fact that while e.g. order-three tensor operations can be $O(m^3)$, in any realistic setting, including those in our paper, $m$ is in the low double digits at most. Coordinate descent can be naively non-trivial with such values of $m$, but can be sped up by exploiting tricks specific to binarization (e.g. one only needs to consider parameter values in the empirical support of each verifier). It is further worth noting that if one picks a continuous transformation family (e.g. sigmoid), gradient descent can be used in place of coordinate descent.

---

[4]The intuition that binarization is strictly harmful because it 'loses information' is incorrect, however, as binarization can improve the decisiveness of a verifier signal.

# E. Experimental details

Several of our experiments involve generation of responses and verifications from open-source models. All such generation was done through vLLM 0.13.0 (Kwon et al., 2023) on a compute node with 8 NVIDIA H100 GPUs (80 GB memory each). We use the following sampling parameters:

- `temperature`: For generation of repeated responses, we set the temperature to be 1.0 by default. Alternative values were used if recommended by the HuggingFace model card. For verifications, a temperature of 0.0 was used for replicability. Note that this does *not* affect notions of conditional independence like TCI, since there is randomness in responses not accounted for by $y(q, r) = \pm 1$ even when verifier signals are deterministic.

- `top_p`: 0.95 by default. Alternative values were used if recommended by the HuggingFace model card.

- `max_model_len`: The maximum possible value for each model.

It is also necessary to extract ground truth correctness labels for the responses that we generate. For multiple choice questions, we deterministically parse the tagged final answer. For short answer questions, we used Qwen3-Next-80B-A3B-Instruct to compare the tagged final answer to a ground truth solution. Appendix E.3 contains prompts for this ground truth extraction pipeline, as well as for generation and verification. We hand-audited a large fraction of responses (around 25%) to check the automated ground truth extraction and evaluation procedure.

**Tie-breaking** In our test-time scaling experiments, a single response must be selected out of $K$ candidates. However, many selection rules (e.g. picking the response with the largest logit) can produce ties. In such cases, we record the accuracy of the selection as the fraction of tied responses which are correct.

## E.1. Baselines

All experimental settings implement and report the following baselines.

- **Pass@1** This baseline simply returns the first response $r_1$ to a query. It requires neither repeated sampling nor verification.

- **Pass@k** This baseline involves generating $k$ responses $r_1, \ldots, r_k$ for each query, and deeming a query solved if at least one response is correct (Chen, 2021). This baseline requires knowledge of ground truth labels and does not involve verification.

- **Majority vote** Given $k$ responses $r_1, \ldots, r_k$ to a query, majority vote returns the most common response $r^\star$. This baseline requires neither ground truth labels nor verification.

- **Naive ensemble** All verifiers are weighted equally, so the score of each response is the average of its post-normalization verifier scores. This method requires verifiers but is unsupervised. To ensure scores produced by different verifiers are comparable, we normalize each verifier's scores to $[-1, 1]$ using min-max normalization.[5]

- **WEAVER (Saad-Falcon et al., 2026)** This semi-supervised baseline uses a held-out set of queries with ground truth labels to estimate $\mathbb{P}(y_i = 1)$ and to adaptively binarize and drop verifiers. The 'inner loop' that provides parameter estimates is gradient descent on a method-of-moments objective that assumes joint conditional independence.

- **Logistic Regression** This semi-supervised baseline involves fitting a logistic regression using ground truth labels for 5% of queries and verifier outputs as covariates. That is, we fit $\beta_0, \beta$ in the model $P(y_{ik} = 1) = \sigma(\beta^T (\mathbf{V}_k)_{i\bullet} + \beta_0)$ using the defaults in scikit-learn.

- **Naive Bayes** This semi-supervised baseline assumes conditional independence, and selects the response with the largest value of

$$\frac{\mathbb{P}(y_i = 1 \mid \mathbf{V}_{i\bullet})}{\mathbb{P}(y_i = -1 \mid \mathbf{V}_{i\bullet})} = \frac{\prod_j \mathbb{P}(v_{ij} \mid y_i = 1)\mathbb{P}(y_i = 1)}{\prod_j \mathbb{P}(v_{ij} \mid y_i = -1)\mathbb{P}(y_i = -1)},$$

---

[5]Concretely, given scores $v(q, r_1), \ldots, v(q, r_N)$ for responses to a query, we map each score via $x \mapsto 2 \times \frac{x - \min_i v(q, r_i)}{\max_i v(q, r_i) - \min_i v(q, r_i)} - 1$

| Method | Requires Verifiers | Supervised | Oracle | Question-Conditional |
|---|---|---|---|---|
| Pass@1 | No | No | No | Yes |
| Pass@k | No | Yes | Yes | Yes |
| FUSE | Yes | No | No | Yes |
| WEAVER (Saad-Falcon et al., 2026) | Yes | Yes | No | No |
| Naive Ensemble | Yes | No | No | Yes |
| Majority Vote | No | No | No | Yes |
| Logistic Regression | Yes | Yes | No | No |
| Naive Bayes | Yes | Yes | No | No |
| Oracle Best Verifier | Yes | Yes | Yes | Yes |

*Table 5.* Comparison of methods and requirements.

where all probabilities on the RHS are estimated from 5% of queries that have ground truth labels. As this baseline assumes binary scores, we employ median binarization—the top 50% of each verifier's scores are mapped to 1, while the remainder are mapped to -1.

- **Oracle Best Verifier** The verifier with the highest balanced accuracy.

For convenience, we collect the requirements of each baseline method in Table 5.

### E.2. Experiments on datasets from Saad-Falcon et al. (2026)

We use the data available at `https://huggingface.co/collections/hazyresearch/weaver` with zero modifications. Therefore, our setting consists of:

- **Benchmarks** GPQA, GPQA Diamond, MATH500, and subsamples of MMLU and MMLU Pro. The MMLU subsample consists of all college-level biology, chemistry, physics, mathematics, computer science, and medicine questions. The MMLU Pro subsample is 500 randomly chosen questions out of 12,000.

- **Generator models** Llama 3.1 8B Instruct and Llama 3.3 70B Instruct.

- **Verifier models** 33 open-source reward models and binary LM judges, which output real-valued and binary scores respectively. When the generator model is Llama 3.1 8B Instruct, all binary LM judges are excluded. See Table 6

- **Number of generations per question** 100.

Running our experiment on this data solely consists of parsing it into matrices before computing the performance of FUSE and baselines. To obtain numbers for WEAVER, we run the replication code available at `https://github.com/HazyResearch/scaling-verification`[6]. All other baseline numbers are manually implemented, with semi-supervised methods allowed to use labels from 5% of questions per benchmark.

The verifiers in this dataset are correlated given the true response: see Figure 4 for the average conditional correlations in this data (conditional correlations weighted by label frequency). In particular, the correlations between score-based reward models are quite large, showing both positive and negative correlations. This is the case both for correlations given correct responses and given incorrect responses.

### E.3. Prompts for generation, verification, and ground truth extraction

We used variants of the following two prompts (the first was adapted from Saad-Falcon et al. (2026)) to generate responses to a given question query.

---

[6]This replication code reveals several typos in Tables 1 and 3 of Saad-Falcon et al. (2026), which we amend in Table 1 and related figures.

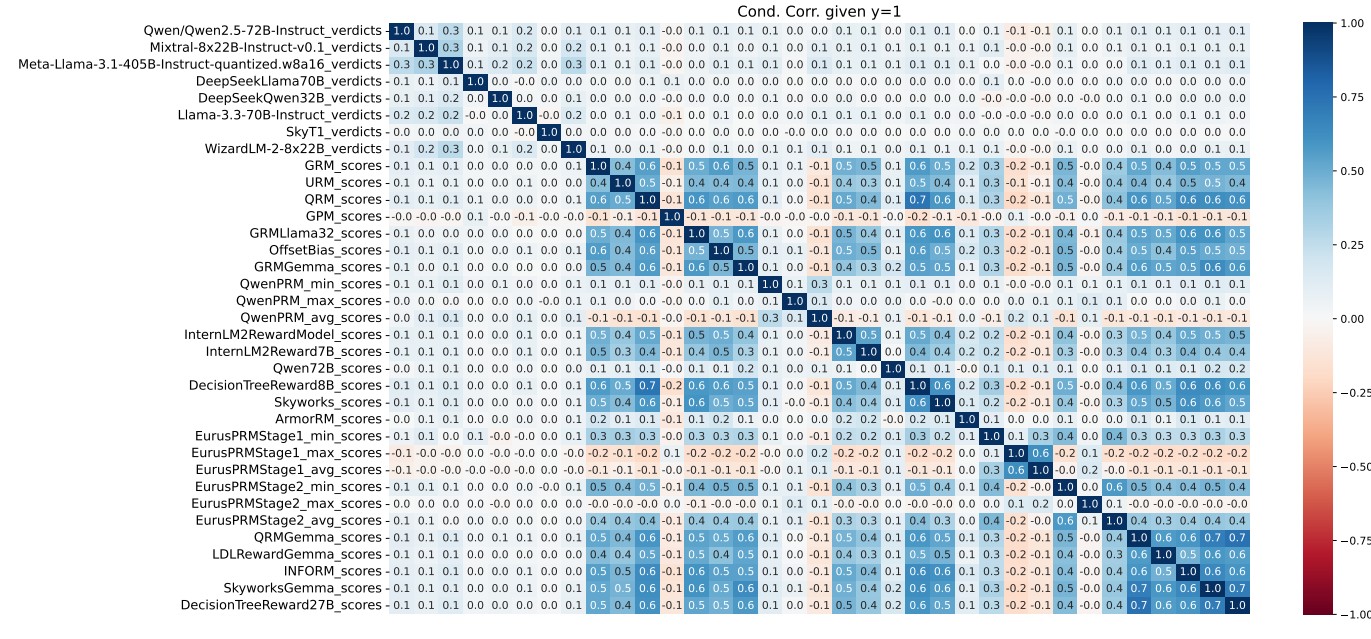

*(a)* Correct response ($y = 1$).

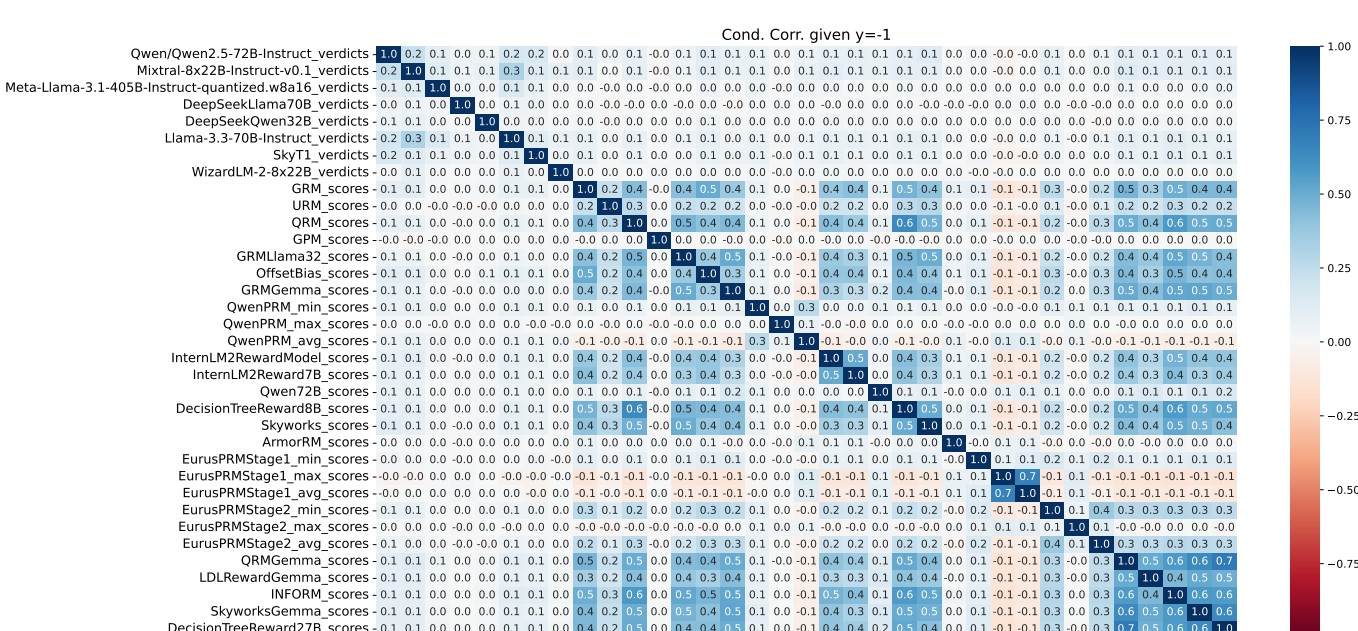

*(b)* Incorrect response ($y = -1$).

*Figure 4.* Average conditional correlations in MMLU-Pro data based on model verdicts and scores for (a) correct responses ($y = 1$) and (b) incorrect responses ($y = -1$).

```
Your response should be in the following format:
Explanation: {your explanation for your answer choice}
Answer: {your chosen answer}
Confidence: {your confidence score between 0% and 100% for your answer}
```

Example generation prompt for IMO Shortlist

```
Your task is to answer the following question:

```
{question}
```

Think step by step. Enclose your final result within a pair of `<answer>` tags
without any additional formatting.

Your response should look like this:

```
{{thinking}}
<answer>
{{your_answer_here}}
</answer>
```
```

The following prompt was adapted from Saad-Falcon et al. (2026) for evaluation of extracted short-answer solutions against ground truth. For multiple choice questions, answers were instead deterministically parsed.

Example evaluation prompt

```
    Compare the following solutions to the given problem and determine
    if they are equivalent. Return True only if the solutions are equivalent.

    Solutions are short response, and may include mathematical expressions.
    You should carefully check for semantic equivalence.
    For example, 'One-half' and '\\frac{{10}}{{20}}' are equivalent.

    Solution 1:
    {extracted}

    Solution 2:
    {ground_truth}

    Enclose your final verdict as : <verdict>VERDICT</verdict>.
    For example, if True, write <verdict>True</verdict>.
```

The following is a sample prompt used for verification generation given a sample response and the original question query. We modified prompts for each verifier model to adapt to its capabilities and idiosyncrasies.

Example verification prompt

```
You are a strict auditor for technical tasks (math/science/coding/logic).

You get a problem and a candidate solution.
Score correctness WITHOUT any ground-truth.

GOAL: minimize false positives. Assume the solution is wrong until proven
correct. Do NOT give benefit-of-the-doubt.

Only reward what you can independently confirm from the problem.

Process (must follow):
1) Extract the FINAL ANSWER (last clearly committed result). If
ambiguous/conflicting -> NOT VERIFIED.
2) Independently check it: do at least ONE concrete verification step
(re-derive a key equation, test a case, check units, run through logic, etc.).
If you cannot perform a real check -> NOT VERIFIED.
3) Decide VERIFIED only if your check(s) confirm the final answer.

Hard constraints:
- Scores 4 or 5 are ONLY allowed if VERIFIED.
- If NOT VERIFIED, score MUST be <=3 (and usually 2 unless strong partial
progress).
- If you find a fatal flaw or the final answer is wrong -> score in {0,1,2}.
- If uncertain, choose the LOWER score.

Rubric (0-5):
5 = VERIFIED final answer; reasoning sound/complete; no meaningful gaps.
4 = VERIFIED final answer; reasoning has gaps/mistakes but answer still correct.
3 = NOT VERIFIED, but strong partial progress + multiple correct key steps;
close to verifiable.
2 = Some correct ideas, but major gaps/errors OR cannot justify final answer.
1 = Mostly wrong; minimal relevant correctness.
0 = Non-solution/irrelevant/refusal/nonsense.

Output format:

1) Brief analysis that begins with 'VERIFIED' or 'NOT VERIFIED'
and mentions your concrete check.

2) New line at end: <score>X</score> where X is 0..5. Nothing after </score>."

Problem:
```
{query}
```

Candidate Solution:
```
{generated responses}
```
```

```
Verify the solution as best you can from the problem.
Then output the final score as <score>X</score>.
```

### E.4. Experiments on Humanity's Last Exam

This experimental setting involves generation of responses and verifications through the Google Gemini API, OpenAI API, and DeepSeek API in addition to local generation using open-source models.

- **Benchmark** A subsample of Humanity's Last Exam. As many verifiers do not natively support multi-modal input, we first removed 394 questions requiring multi-modal input from the 2477 total available at `cais/hle-rolling` on HuggingFace. A batch API request for 100 responses per question from Gemini-3-pro-preview was then submitted with the following sampling parameters: `temperature = 1.0`, `top_p = 0.95`, and a token limit of 30,000 per response. We then obtained our final sample by taking the first 50 responses from each question with at least 50 successful responses and at least one correct response (649 total; 181 multiple choice, 468 exact match). A notable feature of this benchmark is its coverage of a wide range of topics ranging from more standard subjects such as mathematics to fields in the humanities such as dance and literature. See Table 7 for a categorical breakdown of the subsample, where we use the categories defined in the original HuggingFace dataset `cais/hle-rolling`.

- **Generator model** Gemini 3 Pro Preview[7]

- **Solution extraction model:** Qwen3-Next-80B-A3B-instruct

- **Verifier models**

  - Qwen2.5-72B-Instruct
  - Skywork-Critic-Llama-3.1-70B
  - gpt-oss-120b (no Harmony response format[8])
  - Gemini 3 Flash Preview
  - DeepSeek-V3.2
  - GPT-5.2 (high reasoning)
  - GPT-5 mini (high reasoning)

- **Number of Generations per Question** 50

The pooled conditional correlation in this data (computed on all pooled solutions, conditional on the ground-truth label; thus effectively weighted by label frequency) are shown in Figure 5. We observe positive correlations among verifiers with two main blocks: (1) two weaker open-source models Qwen2.5-72B-Instruct and Skywork-Critic-Llama-3.1-70B; and (2) four stronger models gpt-oss-120b, DeepSeek-V3.2, GPT-5.2 (high reasoning), and GPT-5 mini (high reasoning).

We impute missing rubric scores as 0 (i.e. verifier identifies the response as fully incorrect). Across the seven verifier models, 8.63% of scores are missing, primarily due to batch API call failures, safety-related refusals/flags, and length or context-window constraints.

### E.5. Experiments on IMO Shortlist

All data for this experiment were generated from open-source models.

- **Benchmark** The IMO Shortlist subset of IMO AnswerBench (Luong et al., 2025). This 123-question benchmark consists of modified versions of past problems in the IMO Shortlist. The modification, which is done by experts, helps avoid memorization. See Table 8 for a categorical breakdown of IMO Shortlist subset, based on the standard categories of Algebra, Combinatorics, Geometry, and Number Theory.

---

[7]As Humanity's Last Exam is substantially more difficult than conventional benchmarks, our primary consideration in generator selection was ensuring we had a model capable of getting non-trivial pass@k values. At the time of experimentation, Gemini 3 Pro had the best reported pass@1 score of any publicly available model (closed or open), motivating our choice.

[8]See https://developers.openai.com/cookbook/articles/openai-harmony. Interestingly, we found that omitting Harmony slightly improves verification quality on ultra-hard tasks.

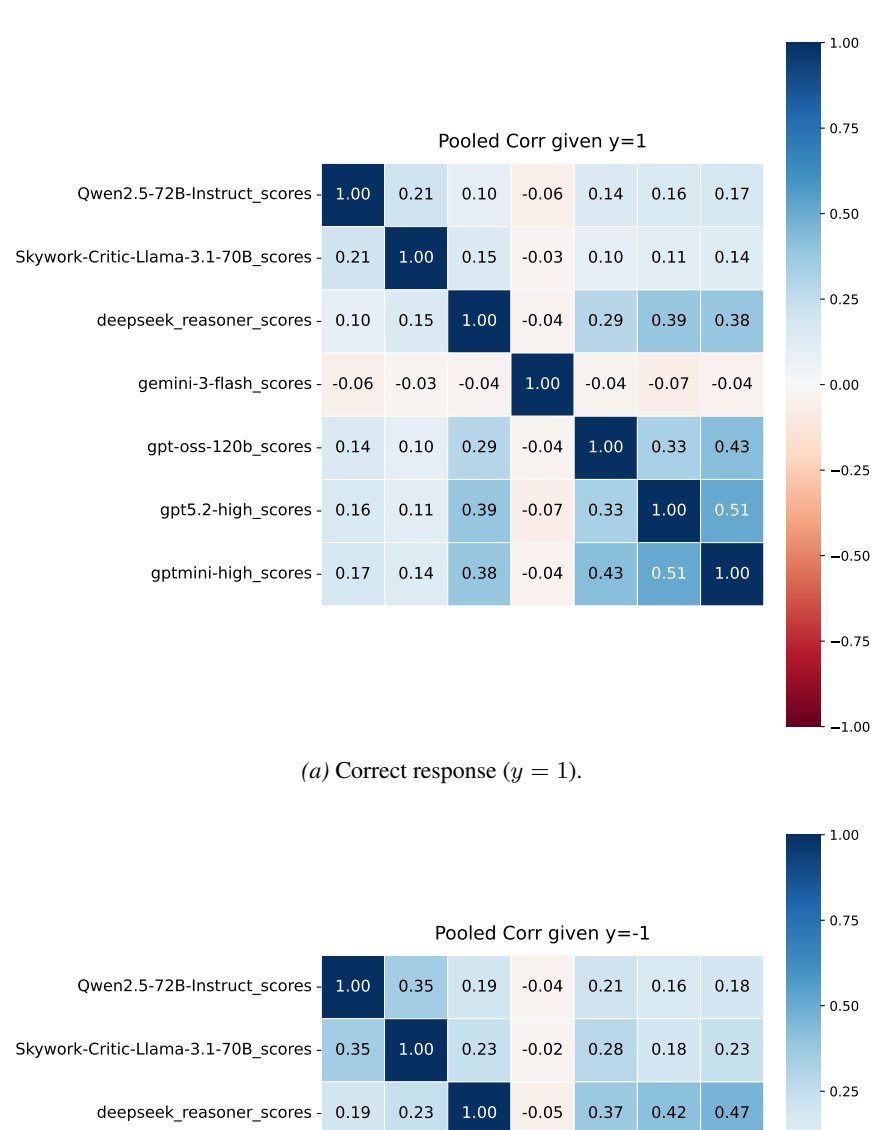

*(a)* Correct response ($y = 1$).

*(b)* Incorrect response ($y = -1$).

*Figure 5.* Pooled correlations in HLE data conditional on (a) correct responses ($y = 1$) and (b) incorrect responses ($y = -1$). Raw scores are used without normalization or binarization.

- **Generator model** Qwen3-30B-A3B-Thinking-2507[9]

- **Solution extraction model** DeepSeek-R1-Distill-Llama-70B

- **Verifier models**

  - DeepSeek-R1-Distill-Qwen-32B
  - Kimi-Linear-48B-A3B-Instruct
  - Llama-3.3-70B-Instruct
  - Ministral-3-14B-Reasoning-2512
  - Ministral-3-8B-Instruct-2512
  - NVIDIA-Nemotron-3-Nano-30B-A3B-BF16
  - Qwen3-30B-A3B-Thinking-2507
  - gemma-3-27b-it
  - gpt-oss-20b

- **Number of generations per question** 50

The average conditional correlation in this data (conditional correlations weighted by label frequency) are uniformly mild and positive (see Figure 6). Further, verifier balanced accuracies are, with a small exception in gpt-oss-20b, homogeneous and only slightly better than random (see Figure 7).

We impute missing rubric scores as 0 (i.e. verifier identifies the response as fully incorrect). Across the nine verifier models, 4.15% of scores are missing, primarily due to length or context-window constraints.

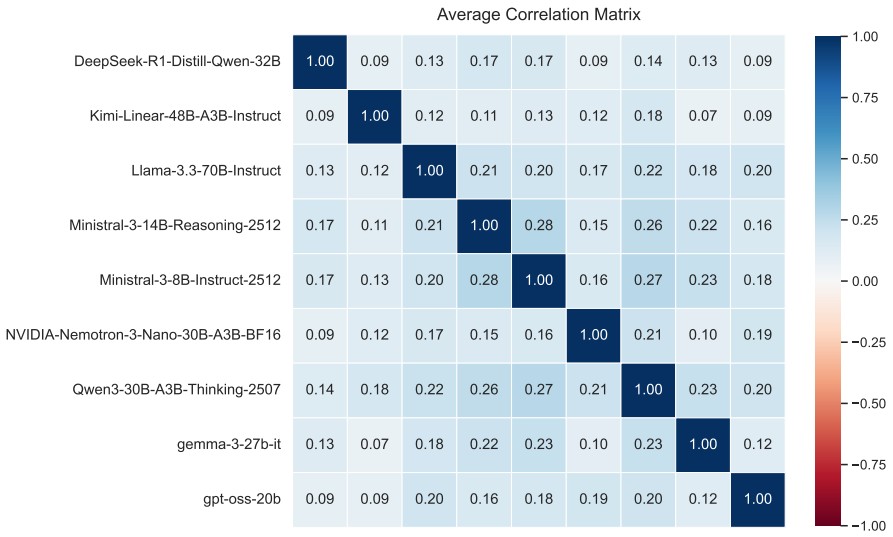

*Figure 6.* Expected conditional correlations of the verifiers given response correctness averaged over all responses (i.e. $(i, j)$th entry is $\text{corr}(v_i, v_j | y = 1)p(y = 1) + \text{corr}(v_i, v_j | y = -1)p(y = -1))$ in IMO Shortlist data. Verifier scores are used without normalization or binarization.

### E.6. Mixed data ablation

All data for our mixed data ablations are from Saad-Falcon et al. (2026). As in our main experiments, we make no modification to either the raw data or the verifier ensemble.

---

[9]We chose this generator model as we expected it to attain non-trivial pass@k values while being runnable on our GPU resources.

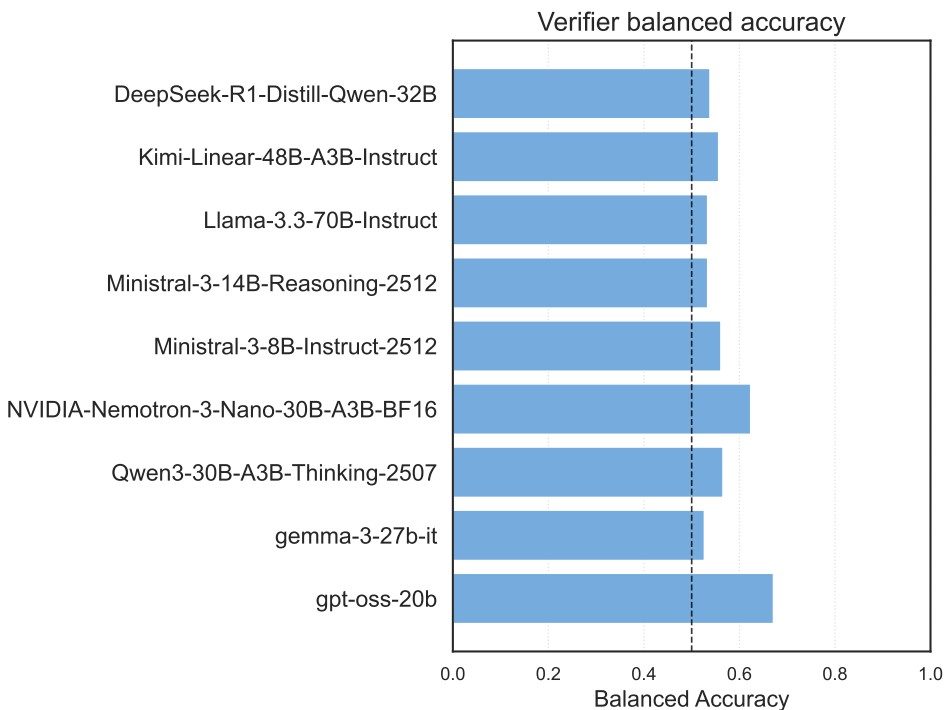

*Figure 7.* Balanced accuracies of verifiers on IMO Shortlist data.

### E.7. Repeated verification is redundant

Let $v_j(r_i)$ denote i.i.d verifications of response $r_i$. When verifications are i.i.d, the optimal ensemble is a naive ensemble, so the induced selection rule is the following:

$$i^\star = \arg\max_i \frac{1}{m} \sum_{j=1}^{m} v_j(r_i).$$

Observe that for any $m$, by linearity of expectation, the naive ensemble $\frac{1}{m} \sum_{j=1}^{m} v_j(r_i)$ has the same sensitivity, specificity, and balanced accuracy as any individual $v_j(r_j)$. On the other hand, as $m \to \infty$, the above selection rule converges to $i^\star = \arg\max_i \mathbb{E}[v_j(r_i)]$. Scaling repeated verifications therefore simply replaces a stochastic verifier with a deterministic equivalent that has identical balanced accuracy. While balanced accuracy and selection accuracy are distinct, as the former is not sufficient to determine the latter, this explains the observation by Saad-Falcon et al. (2026) that sampling five times from a strong verifier typically performs identically to sampling merely once (see their Table 21).

### E.8. Additional unsupervised baselines

In the main text, we focus on majority vote and naive ensemble as unsupervised baselines as these are the most commonly used methods in the language model setting. Indeed, we are unaware of any prior work that connects the literature on unsupervised ensemble learning to test-time scaling. In Table 9, we report the performance of existing unsupervised baselines on the Saad-Falcon et al. (2026) data. These are:

- Dawid & Skene (1979): An Expectation-Maximization algorithm that assumes conditional independence.

- Jaffe et al. (2016): An extension of Jaffe et al. (2015) to settings with structured conditional dependence.

- Gaussian mixture model: We assume that conditional on the binary label $y_i$, verifier scores are jointly multivariate Gaussian and estimate parameters through the EM algorithm.

These numbers uniformly fall below those of FUSE in Table 1, indicating that our performance is not a mere consequence of recognizing that the unsupervised ensembling literature has relevance to LLM verification.

### E.9. Ensemble size robustness

We investigate the performance of FUSE and competing baselines when the number of available verifiers is very small. We consider the Saad-Falcon et al. (2025) 8B setting by randomly sampling five verifiers from the original set of 19. The best-of-100 accuracies are reported in Table 10. Using only five verifiers appears to decrease the overall accuracy of FUSE (resulting in a larger gap to Pass@100), but other baselines also see changes in accuracy so FUSE remains competitive overall. Naive ensemble does surprisingly well here vis-a-vis essentially all baselines (except for MATH500), which may reflect its robustness.

## F. Compute requirements

Verifications by non-API models for Humanity's Last Exam and all responses and verifications for IMO AnswerBench were generated locally on a compute node with 8 NVIDIA H100 GPUs with 80 GB of memory each. All other aspects of this work, including the experiments on data from Saad-Falcon et al. (2026), did not require GPU usage or other forms of specialized compute.

## G. Replicability

We open-source our raw HLE and IMO Shortlist data at `https://huggingface.co/FUSE-verifiers`. Our supplementary material contains instructions for running the replication code of Saad-Falcon et al. (2026), which we use to generate the Weaver numbers in Table 1. Code is available at `https://github.com/FUSE-verifiers/fuse-replication`.

| Name | Type | 8B | 70B |
|------|------|----|----|
| DeepSeekLlama70B | binary | No | Yes |
| DeepSeekQwen32B | binary | No | Yes |
| Llama-3.3-70B-Instruct | binary | No | Yes |
| Meta-Llama-3.1-405B-Instruct-quantized.w8a16 | binary | No | Yes |
| Mixtral-8x22B-Instruct-v0.1 | binary | No | Yes |
| Qwen/Qwen2.5-72B-Instruct | binary | No | Yes |
| SkyT1 | binary | No | Yes |
| WizardLM-2-8x22B | binary | No | Yes |
| ArmorRM | reward model | Yes | Yes |
| DecisionTreeReward27B | reward model | No | Yes |
| DecisionTreeReward8B | reward model | No | Yes |
| EurusPRMStage1_avg | reward model | Yes | No |
| EurusPRMStage1_max | reward model | Yes | Yes |
| EurusPRMStage1_min | reward model | No | Yes |
| EurusPRMStage2_avg | reward model | No | Yes |
| EurusPRMStage2_max | reward model | Yes | No |
| EurusPRMStage2_min | reward model | Yes | Yes |
| GPM | reward model | Yes | Yes |
| GRMGemma | reward model | Yes | Yes |
| GRMLlama32 | reward model | Yes | No |
| GRM | reward model | Yes | Yes |
| INFORM | reward model | No | Yes |
| InternLM2Reward7B | reward model | Yes | Yes |
| InternLM2RewardModel | reward model | No | Yes |
| LDLRewardGemma | reward model | No | Yes |
| OffsetBias | reward model | Yes | Yes |
| QRMGemma | reward model | No | Yes |
| QRM | reward model | Yes | Yes |
| Qwen72B | reward model | No | Yes |
| QwenPRM_avg | reward model | Yes | Yes |
| QwenPRM_max | reward model | No | Yes |
| QwenPRM_min | reward model | Yes | Yes |
| SkyworksGemma | reward model | No | Yes |
| Skyworks | reward model | Yes | Yes |
| URM | reward model | Yes | Yes |

*Table 6.* Verifiers in Saad-Falcon et al. (2026) experimental setting

*Table 7.* Category summary for HLE subset.

| Category | Count |
|----------|-------|
| Math | 141 |
| Humanities/Social Science | 110 |
| Other | 87 |
| Computer Science/AI | 84 |
| Biology/Medicine | 77 |
| Physics | 77 |
| Chemistry | 48 |
| Engineering | 18 |
| Chess/Logic/Puzzle | 7 |

*Table 8.* Category summary for IMO Shortlist subset.

| Category | Count |
|---|---|
| Algebra | 35 |
| Combinatorics | 45 |
| Number Theory | 35 |

*Table 9.* Best-of-100 selection accuracy of additional unsupervised baselines on Saad-Falcon et al. (2026) data.

| Size | Benchmark | DS | GMM | DCL |
|---|---|---|---|---|
| | GPQA | 0.3168 | 0.3709 | 0.2832 |
| | GPQA Diamond | 0.3295 | 0.4184 | 0.3279 |
| 8B | MATH500 | 0.5513 | 0.6729 | 0.6804 |
| | MMLU | 0.7179 | 0.8032 | 0.7910 |
| | MMLU-Pro | 0.5394 | 0.6408 | 0.6199 |
| | GPQA | 0.5427 | 0.5670 | 0.5601 |
| | GPQA Diamond | 0.5240 | 0.5413 | 0.5606 |
| 70B | MATH500 | 0.8115 | 0.8726 | 0.8693 |
| | MMLU | 0.8480 | 0.8952 | 0.9013 |
| | MMLU-Pro | 0.7497 | 0.8263 | 0.8220 |

*Table 10.* Best-of-100 accuracies for all baselines on 8B data from Saad-Falcon et al. (2025), averaged over 20 random subsets of 5 verifiers (from 19 total). Parentheses denote standard errors where relevant.

| | Unsupervised | | | | Supervised | | | | Oracle |
|---|---|---|---|---|---|---|---|---|---|
| Benchmark (8B) | Pass@1 | Majority Vote | Naive Ensemble | FUSE | OBV | Weaver | Logistic | Naive Bayes | Pass@100 |
| GPQA | 26.5 | 30.5 | 37.8 (0.63) | 37.0 (0.68) | 38.6 (0.76) | 35.1 (0.52) | 34.9 (0.76) | 37.3 (0.74) | 95.2 |
| GPQA Diamond | 27.3 | 32.3 | 39.8 (0.88) | 39.8 (1.21) | 38.6 (0.59) | 33.9 (0.48) | 33.9 (1.36) | 39.9 (1.58) | 94.9 |
| MATH500 | 51.0 | 69.6 | 73.9 (1.39) | 75.6 (1.37) | 75.1 (1.26) | 75.3 (1.90) | 79.8 (1.29) | 71.9 (0.92) | 99.2 |
| MMLU | 61.5 | 72.7 | 83.3 (0.45) | 82.4 (0.69) | 82.8 (0.44) | 78.7 (0.77) | 83.0 (0.76) | 80.3 (0.74) | 98.5 |
| MMLU Pro | 44.8 | 56.4 | 67.2 (0.86) | 67.0 (0.80) | 61.8 (0.82) | 60.9 (1.02) | 62.8 (0.71) | 64.4 (0.71) | 96.8 |

*Table 11.* Best-of-100 selection accuracy of FUSE on 8B Saad-Falcon et al. (2026) data when using cross-entropy in lieu of averaged accuracy for predictor optimization.

| Size | Benchmark | CE Accuracy |
|---|---|---|
| | GPQA | 40.5 |
| | GPQA Diamond | 41.1 |
| 8B | MATH500 | 83.6 |
| | MMLU | 84.9 |
| | MMLU-Pro | 67.0 |

