# OpenReview forum: "FUSE: Ensembling Verifiers with Zero Labeled Data"
_ICML.cc/2026/Conference — ICML 2026 regular_

### Official Review · Reviewer_Wv7k · 2026-03-08

**Soundness:** 3
**Presentation:** 3
**Significance:** 3
**Originality:** 2
**Overall Recommendation:** 5
**Confidence:** 4

**Summary:**

This paper focuses on the setting where we have access to multiple verifiers, multiple responses to a prompt, and wish to select the best response. However, we also have access to *zero* labeled examples, which makes standard ensembling techniques difficult. The authors adapt ideas from Jaffe et al (2015), which they discuss in detail. Unlike Jaffe et al, however, this paper does not assume the verifiers are conditionally independent (conditionally independent given the true label). This is an important distinction, as models/verifiers often make  mistakes on the same examples. The authors develop a heuristic to test how far from joint conditional independence the verifiers are. This heuristic will be zero under triplet conditional independence. The authors are also able to adapt to continuous verifiers by binarizing the verifiers at the threshold that minimizes this heuristic.

For ensembling, the authors first estimate the sensitivity and specificity of the verifiers and use those estimates to obtain a posterior over the true label. The authors then write down a linear objective as a function of the verifier scores and these estimated posteriors and maximize it.

The authors experiment on several datasets. Surprisingly, the method outperforms even supervised baselines. The method particularly shines on hard problems.

**Compliance With Llm Reviewing Policy:**

Affirmed.

**Final Justification:**

I believe this is a strong paper and builds well on prior work from Jaffe et al. I recommend acceptance.

**Key Questions For Authors:**

Q1: Can you discuss any connections to conditional independence testing? For example, see https://www.pywhy.org/pywhy-stats/v0.1/conditional_independence.html. One view of Proposition 2.4 could be a form of conditional independence testing. However, in your setup there is no "null" distribution of this metric, so it is hard to say how much this summed variance must exceed zero in order to be significant.

Q2: Why does your method beat even supervised methods? This is quite surprising. Please add discussion.

Q3: Could you please clarify how to recover b from the marginal moment tensors (Section 2.1)?

Q4: Could you also clarify what you mean by "Plugging these estimates into (4) yields an empirical approximation of (4)"?

**Limitations:**

Yes.

**Strengths And Weaknesses:**

Strengths:
(S1) The paper is well-written and easy to follow. Notation is clearly defined, and the authors go into sufficient depth on Jaffe et al (2015) while also crediting those authors.

(S2) Empirically, the method beats other methods across three data sources and two different models. The experiments importantly span a wide range of numbers of verifiers. One experiment uses seven verifiers, while another uses 33.

(S3) The heuristic in proposition 2.4 is well motivated and justifies using it as a test for non-independence.

Weaknesses:
(W1) I am not sure where the optimization objective in (7) came from. There are many linear objectives one could write, and I'm not sure why this one is preferred.

(W2) There is a general issue of hypothesis testing. In particular, the authors propose a metric to quantify TCI violation. However, this metric only has to equal zero in expectation if TCI holds. Even when TCI holds, on any given finite sample, it may be violated. How can you construct a "null" distribution of the strength of TCI violations?

---

> ### Author Rebuttal · Authors · 2026-03-31
>
> We thank the reviewer for their thoughtful and encouraging review, especially for saying that our paper is well-written and for calling our TCI heuristic well-motivated. Below, we address the reviewer’s questions and concerns.
>
> **W1:** The objective loss arises as an approximation of average accuracy given our estimates of verifier sensitivity and specificity. Specifically, Eq. (7) serves as a proxy for the quantity in (6), which measures ${\sum_i \mathbb{E}[Y(q,r_{i})\mid V] \hat{f}\_{\theta}(V_{i\bullet\})}$, where the expectation is taken over response correctness given the observed verifier scores (i.e., randomness in $r$ given $V$). Other losses may be more appropriate depending on the function class chosen for step 3 of FUSE. For example, in logistic regression where $f_{\theta}$ represents the model's probability of $y = 1$, the average cross-entropy $\sum_i p^\star(r_i) \log f\_{\theta}(V_{i \bullet}) + (1-p^\star(r_i)) \log (1-f\_{\theta}(V_{i \bullet})),$ with $p^\star$ approximated by $\hat{p}$, may be a more natural objective. We adopt average accuracy here as a general-purpose metric, though the framework readily accommodates alternative losses. We will provide a more in-depth empirical ablation in the appendix of the revised paper comparing average cross-entropy loss with average accuracy across our experiments.
>
> **W2:** We thank the reviewer for raising this point. We agree that the TCI statistic is subject to finite-sample fluctuations, and want to be precise about how we use it. Rather than as a test statistic for hypothesis testing, we use it as a heuristic optimization objective: we find transformations $g$ of the score matrix that minimize the statistic, with the goal of reducing empirical violations of TCI. This is somewhat analogous to how empirical risk is used to train predictive models in lieu of the unavailable population risk. We will add a clarifying note of the TCI statistic’s role upon its introduction.
>
> **Q1:** Thank you for this question. This ties into our discussion for W2. In particular, we use the TCI statistic not to test a hypothesis with Type I error control, but as an optimization objective for finding transformations of the score matrix that bring it closer to satisfying the TCI assumption. Regarding the suggested tests, while these may be useful for gauging conditional independence in other settings, they require access to observed labels, which are unavailable in our setting.
>
> **Q2:** This is a good question, and one we also found initially puzzling. The reasons vary depending on which baseline method is being compared against. For baseline models like Naive Bayes and Weaver that aggregate signals based on the assumption that the verifiers are jointly conditionally independent given the true labels, we believe that the decline in performance is due to the model misspecification (the conditional correlation plots demonstrate verifiers are indeed often very conditionally correlated) while FUSE seeks to be robust to these correlations. For the oracle best verifier (OBV), relying solely on the single most accurate verifier discards potentially useful signals from the remaining ones. For logistic regression, we hypothesize that the lower performance is driven by high variance due to the small training sample size (only 5% of each benchmark). We will add these comments in Section 3.1 under the discussion for “FUSE competes well with semi-supervised baselines.”
>
> **Q3:** We follow the procedure presented in Jaffe et al (2015). In particular, note that since $\mathbf{u} = \sqrt{1-b^2}(2\mathbf{\pi} -1)$ (Eq (1)) and $\mathbf{w} = (-2b(1-b^2))^{1/3} (2\mathbf{\pi} -1 )$, $\mathbf{T} = \alpha(b) \mathbf{u} \otimes \mathbf{u} \otimes \mathbf{u}$ on the off-diagonal, where $\alpha(b) := -2b/(1-b^2)^{1/2}$. Using this relation, we first obtain $\hat{\mathbf{u}}$ from $\hat{\mathbf{\Sigma}}$, then obtain an estimate $\hat{\alpha} := \arg\min\_{\alpha} \sum\_{j\_1 < j\_2 < j\_3} (\hat{T}\_{j\_1, j\_2, j\_3} - \alpha \hat{u}\_{j\_1} \hat{u}\_{j\_2} \hat{u}\_{j\_3})^2$, and finally solve for $b$ via $\hat{b} = -\frac{\hat{\alpha}}{\sqrt{4 + \hat{\alpha}^2}}$. We will reiterate this construction for completion and clarity in the Appendix.
>
> **Q4:** Since the left-hand side of Eq. (4) is defined in terms of population covariance tensors, we replace $\Sigma$ and $T$ with empirical estimates computed from the transformed score matrix $g(V)$. We then form the ratios $\hat T\_{j\_1,j\_2,j\_3}/\hat\Sigma
> \_{j\_1,j\_2}$​​, compute for each fixed $j\_3$​ the sample variance of these ratios over $1\leq j_1 < j_2 < j_3$, and sum over $j_3$​. The resulting quantity, $\hat S(g(V))$, is an empirical approximation to the left-hand side of (4) and provides a feasible measure of deviations from TCI among the transformed verifiers. We will clarify: “Plugging in empirical estimates of the second and third-order moment tensors of $g(V)$ into (4) yields an empirical approximation of (4).”

---

> > ### Author Rebuttal · Reviewer_Wv7k · 2026-04-02
> >
> > Thank you for the additional comments. I will maintain my "accept" score and encourage the authors to include a brief paragraph discussing the comparison to supervised baselines.

---

### Official Review · Reviewer_czEZ · 2026-03-12

**Soundness:** 3
**Presentation:** 3
**Significance:** 3
**Originality:** 3
**Overall Recommendation:** 5
**Confidence:** 3

**Summary:**

This paper proposes FUSE, a method to ensemble multiple (oracle-free) verifiers. In high-level. fuse take three steps (1) transform verifier scores to be TCI-compatible, (2) estimate each validator's quality (3) estimates the posterior correctness probablities.

In short, this is an effort to statisically put together scores of weak verifiers (without labels) and learn a better ensemble outperforming naive averaging or majority vote.

As shown in Figure 4 it seems like the authors leverage 33 models ranging from 8B RMs to 72B or 8x22B LLM-Judges for individual scores and put them together. The paper uses a wide array of sampled benchmarks including GPQA, MATH500, MMLU, MMLU-Pro, IMO shortlists and HLE. Generally FUSE outperforms all unsupervised methods and scores near supervised best models showing impressive performance.

**Compliance With Llm Reviewing Policy:**

Affirmed.

**Final Justification:**

The response from the authors have handled my concerns.

**Key Questions For Authors:**

see weakness

**Limitations:**

yes

**Strengths And Weaknesses:**

Generally I think the paper is well written with clear writing and strong experiments. I have two concerns (or suggestions)

1) i think it will necessary to include an analysis on the cost of these verification methods. Suggested FUSE requires evaluation from 33 different verifiers, making it very costly. an analysis on how cheap it can be (whats the minimum number of models that makes this possible) seems important for the method to be used in real life.

2) the model mostly uses two main generators Llama-3.1-8B and 70B for the experiments. Qwen3-30B is used for IMO shortlist and Gemini-3-Pro for HLE I think there should a clarification on how the models were selected.

3) Additionally it seems like the benefit of FUSE seems to saturate as the difficulty of the questions climb. for instance, FUSE outperforms most methods by a big margin in Table 1. the gap is smaller in table 3. the gap is even smaller (or is caught up by naive ensemble) in table 4. why is this so? is the method more effective in the easier range? or does this have to do with the capability of verifiers?

---

> ### Author Rebuttal · Authors · 2026-03-31
>
> We thank the reviewer for their detailed assessment and constructive feedback. We also thank the reviewer for saying that our paper is “well written with clear writing and strong experiments.” In response to specific comments:
>
> **Cost analysis:**
> Mathematically, our method requires a minimum of three verifiers to be well-defined. Our Table 1 experiments use the setting of Saad-Falcon et al (2025), and hence have 16 verifiers for 8B and 32 for 70B. In the HLE and IMOBench experiments, however, we use far fewer verifiers -- seven and nine respectively. We will emphasize this in the paper.
>
> It’s nonetheless a good idea to empirically investigate the performance of FUSE and competing baselines when the number of available verifiers is very small. We do so in the 8B context by randomly sampling 5 verifiers from the original set of 16*. More extensive ablations will appear in the Appendix of future versions. All numbers below are Best-of-100 selection accuracy.
>
> |  | Pass@1 | Maj Vote | Naive Ens | FUSE | OBV | Weaver | Logistic | Naive Bayes | Pass@100 |
> |---|-------|---------|---------|-------|---------|---------|---------|---------|---------|
> GPQA | 28.3 |  30.5   | 36.8 |  37.2 |  40.0  | 33.2 | 34.2 | 33.7 | 95.2 |
> GPQA Diamond | 28.3 |  32.3  | 36.7 |  39.4 | 34.7 | 33.3 | 29.8 | 27.9 | 95.0 |
> MATH500 | 49.9 |  69.6  |  74.7  | 77.4 | 77.5 | 77.7 | 82.1 | 75.0 | 99.2 |
> MMLU | 64.1 | 72.7 | 80.8 | 81.6 | 83.8 | 82.8 | 80.8 | 81.1 | 98.5 |
> MMLU Pro | 46.6 | 56.4 | 65.1 | 66.8 | 65.7 | 57.5 | 61.1 | 61.7 | 96.8 |
>
> Using only 5 verifiers appears to mildly decrease the overall accuracy of our method (so a larger gap exists to Pass@100), but as other baselines also see changes in accuracy, FUSE remains competitive overall. Interestingly, variance across benchmarks seems to increase for most baselines -- e.g. logistic convincingly beats all others in MATH500, but is barely above Pass@1 on GPQA Diamond.
>
> (* EurusPRMStage1_min_scores, GRM_scores, InternLM2Reward7B_scores, QwenPRM_avg_scores, QwenPRM_max_scores)
>
> **Selection of generator models:** Our experiments in the Weaver setting (Saad-Falcon et al. 2025) use the data open-sourced by the Weaver authors, so by default have the same generator models (Llama 3.1 8B Instruct and 70B Instruct). For Humanity’s Last Exam, which is much more difficult than conventional benchmarks, our primary consideration was picking a model capable of getting non-trivial pass@k values. At the time of experimentation, Gemini 3 Pro had the best reported pass@1 score of any publicly available model (closed or open), motivating our choice. In the IMOBench experiment, we wanted to use a more current open-source model than Llama 3.1, while also obtaining non-trivial pass@k. To this end, Qwen3-30B was both runnable on our GPU resources and achieved reasonable pass@k values. We will add these clarifications to the relevant subsections in Section 3 of the paper.
>
> **Why the benefits of FUSE seem to saturate as setting difficulty increases:** We believe that on HLE, reduced gains can be attributed to the high difficulty of the benchmark, which induces bimodality in the response distribution -- many questions have very few or nearly all correct responses out of 50, making the Best-of-50 response selection task either trivial or impossible (so methods have less room to distinguish themselves). As an example, 27.89% of questions have either 1 or 50 correct answers in our sample.
>
> Regarding IMO shortlist, we note that, to first order, an optimal ensemble of verifiers depends only on their balanced accuracies (see equation (14) of Parisi et al. and surrounding discussion). Therefore, in settings where all verifiers have similar balanced accuracies and are nearly conditionally independent, which seems potentially the case for the IMO shortlist experiment (see Figures 6 and 7 in the Appendix), we expect the naive ensemble to be close to optimal. The fact that FUSE matches naive ensemble can therefore be viewed as a sign of robustness (as opposed to, e.g., failure to capture signal).
>
> In general, however, it is correct that FUSE's performance--and that of any ensemble method--is naturally bounded by the available signal from the verifiers themselves. We do not believe, and certainly do not wish to claim, that our method is a remedy for settings in which either generation or verification are prohibitively difficult. Nonetheless, it seems likely that potential application areas for FUSE will only grow with time as language model capabilities advance.
>
> As an aside, we note that no unsupervised method can work when we don’t have a majority of verifiers better than random (see discussion of Assumption 3 in Jaffe et al. 2015, pg 3).

---

> > ### Author Rebuttal · Reviewer_czEZ · 2026-04-04
> >
> > Thanks for the response. Ive updated my score to 5.

---

### Official Review · Reviewer_Es1L · 2026-03-13

**Soundness:** 3
**Presentation:** 3
**Significance:** 3
**Originality:** 3
**Overall Recommendation:** 5
**Confidence:** 3

**Summary:**

The paper provides an unsupervised method for (flexibly) ensembling verifiers, with the main motivation of improving test-time scaling. The method builds off innovative proposals from the mid-2010s (Parisi et al. and Jaffe et al.) which use the higher-order moment structure between the verifiers to infer their accuracy (specificity and sensitivity) under a Triplet Conditional Independence (TCI) assumption. Their main modifications are adding a monotonic transformation of the verifiers to reduce a certain violation of the TCI criterion, and a more flexible ensemble construction that appears to work well empirically. Indeed, the paper includes compelling empirical results that show their method substantially improves upon naive unsupervised verifier-ensembling baselines (with the objective of best-of-N-style compute scaling uplift), and can even often perform as well or better than relevant semi-supervised alternatives.

**Compliance With Llm Reviewing Policy:**

Affirmed.

**Key Questions For Authors:**

- Perhaps interesting to consider the relationship to other work that tries to leverage weak supervision with other fundamental intuitions, such as the mutual predictability idea from e.g. https://arxiv.org/pdf/2601.20299. (not necessarily suggesting you need to cite this particular paper, but am curious about this style of idea and wondering if there might be some more related manifestations around?)
- I’d be interested to hear more about whether this method could leverage labelled data (with small adaptation), and be run in a semi-supervised fashion. If so, how would it compare to the existing semi-supervised methods?


Minor criticisms (mostly regarding presentation):
- "it first finds TCI-minimizing transformation" in caption
It is weird wording as TCI is an independence condition when, iiuc, they are minimising the related continuous quantity of violation (4).
Moreover, it might be good to state explicitly that distributions which satisfy (4) do not necessarily satisfy TCI (i.e. (4) is necessary but not sufficient).
- “population covariance tensors between the binarized ver"
Strange to mention binarization before introducing binarization in the following paragraph (iiuc) - perhaps rearrange?
- section 3.1 strange section title - seems more about Saad-Falcon data than the models?
- "FUSE competes well with semi-supervised baselines"
 slightly unnatural wording
- The presentation in Table2 is perhaps a little cheeky in that they bold FUSE when usually worse than a supervised baseline
 (My impression is they should probably also bold the best performing supervised baselines for each task! If so, perhaps underline the unsupervised method when it outperforms all supervised baselines?)
- Somewhat unclear to me what precisely happens with HLE, as I believe some tasks are open-ended; some baselines don’t apply in this setting (what does majority mean here?).
- “Our experiments are comprehensive…” feels like filler.
- “Apply Theorem 2.3 to…” in Algorithm 1 doesn't seem sufficiently precise. It is not clear to me exactly how they are doing this. More details please!
- Lack of uncertainty quantification in empirics. Would be good to have error bars at least.

**Limitations:**

Yes

**Strengths And Weaknesses:**

## Strengths
- The core problem of unsupervised ensembling of verifiers for test-time scaling appears interesting, important, and relatively under-explored
- The core idea of using higher-order moment structure to ‘get at’ signal in this setting is elegant (indeed, the Parisi and Jaffe papers read very well to me).
- The empirical results are strong. (Indeed, somewhat surprisingly so!); and it is nice to see this creative method applied to LLM settings which afaict are interesting/appropriate.
- The paper is generally well presented and relatively easy to read.

## Weaknesses
- The construction is rather involved and has many moving parts.
    - It is not clear to me how easy it would be to get working in practice - and indeed, suspect that certain hyperparameters might be sensitive. There is some discussion in Appendices D and E, but I am not convinced that it would be practical for a user to ‘plug and play’.
    - Relatedly, I have a corresponding mild concern that the authors have ‘tried a load of different things on top of the Parisi/Jaffe idea’ and then ‘picked what looked best’ (raising potential multiple testing issues; see Appendix D for more context; that said Appendix D feels very important so thank you for including this!).
    - Perhaps there are simpler ways to cash out similar underlying intuitions to address the same core problem?
- There is little discussion of the computational cost of the method - and indeed the sort of regimes under which the setting is of practical importance (more below).

A general note: I’m not entirely sure who the audience for this work is. It does not seem super useful out-of-the-box, and I am not sure there are many people looking for a solution to this particular problem - indeed, by default I’d suspect a lab would want to invest test-time compute in using a stronger verifier rather than ensembling weak verifiers (both inference-time or indeed train-time). I’d be interested to hear more about these practical considerations. As a proof-of-concept it is very interesting though.

---

> ### Author Rebuttal · Authors · 2026-03-31
>
> We thank the reviewer for their thoughtful comments and are glad they find the problem of unsupervised ensembling of verifiers to be “interesting, important and relatively under-explored” and our method to be creative. We provide additional remarks below:
>
> > The construction is rather involved...similar underlying intuitions to address the same core problem?
>
> We're sympathetic to these concerns. One view we earnestly hold is that the problem FUSE attempts to address -- unsupervised ensembling without independence assumptions -- is simply very difficult. This justifies some (but certainly not unlimited) complexity.
>
> > Relatedly, I have a…thank you for including this!).
>
> We thank the reviewer for raising this point. This is a common concern for works involving benchmarking, and our paper is certainly not immune to it. That said, we suspect the bias of our reported accuracies is quite small. To further investigate this, we will run FUSE, out of the box, on a new unseen benchmark, if accepted. (Unfortunately, the time costs of response generation and verification are too prohibitive for us to do so during the short rebuttal period.)
>
> > There is little discussion of the computational...As a proof-of-concept it is very interesting though.
>
> We broadly agree that FUSE does not automatically "plug and play" with common existing pipelines for inference, RL, benchmarking, etc. It seems likely that FUSE-type ideas are most valuable when i) verification is particularly important or hard and ii) one has access to a modest inventory of strong but diverse verifiers. By definition, i) will not encompass all scenarios, but we suspect that ii) is rather common. In fact, this may be more true within labs than outside of labs, as labs can access many internal models / checkpoints beyond those publicly accessible.
>
> An additional practical consideration is that the process of iteratively training a model induces constant distribution shift (of relevant prompts, responses, and labels). This weakens the value of both labels that are only periodically collected and static, single verifiers. FUSE is a work-around that, depending on the available verifier set, may incur less overhead than training generator and verifier models in tandem.
>
> Finally, in a rather different direction, one reading of our FUSE $\approx$ semi-supervised results is that existing semi-supervised methods likely have real room for improvement. So, to the extent that one finds such methods worthwhile, our results may be of independent interest.
>
> > Perhaps interesting to consider the relationship to other…e.g. [https://arxiv.org/pdf/2601.20299](https://arxiv.org/pdf/2601.20299)
>
> This is a nice reference – we will discuss it in future versions. One similarity we see is that both works lean on a certain asymmetry. In FUSE, TCI implies the population (not empirical) statistic vanishes, but the converse – a small value of the statistic implies TCI – is not true. Hence, our first-stage optimization is essentially minimizing a heuristic loss. In the linked work, one has with high probability that peer prediction is an ex-ante Bayesian Nash equilibrium. This motivates a mutual predictability-based training reward, which (to our understanding) is also heuristic, in the sense that convergence to equilibrium play is not guaranteed.
>
> Put more plainly, unsupervised learning is hard without strong structure or assumptions. Methods that attempt to circumvent this, by 'fundamental intuitions' or otherwise, will usually require some leap of faith (but may be very effective in practice).
>
> > I’d be interested to hear more about whether this method could leverage labelled data...
>
> One obvious adaptation would be to use labels in estimating quantities like the positive class probability $P(Y=1)$. The empirical results in Saad-Falcon et al (2025) (see Table 4) suggest that simply improving estimation of this parameter can matter a fair deal.
>
> A more subtle question is whether "small adaptation" is appropriate when given access to labels. An interesting feature of Saad-Falcon et al (2025) is that the authors obtain strong results while making a joint conditional independence assumption that is strongly violated in practice. We suspect that this is a consequence of the adaptive binarization step (see Appendix B.3) in which hold-out accuracy is used to search over all possible binarizations of real-valued verifiers. Consequently, they effectively learn a "best mis-specified model", akin to how ordinary least squares typically learns the best linear fit even in the absence of linearity.
>
> Generalizing the above to FUSE suggests that when given labels, we may wish to add an entire "outer loop" for robustness. It is not completely obvious how to do this, as if one wishes to preserve TCI-guided transformation, new parameters must be introduced to hill-climb with.
>
> > Minor criticisms
>
> We thank the reviewer for the detailed comments, and we will incorporate these in the final version.

---

> > ### Author Rebuttal · Reviewer_Es1L · 2026-04-02
> >
> > Thanks for the response. I maintain my acceptance recommendation.

---

### Official Review · Reviewer_qT59 · 2026-03-16

**Soundness:** 4
**Presentation:** 3
**Significance:** 3
**Originality:** 3
**Overall Recommendation:** 4
**Confidence:** 4

**Summary:**

This paper proposes a new approach, FUSE, to ensemble the predictions from multiple verifiers to select most promising response among multiple candidates (i.e., Best-of-N setup). Specifically, FUSE first estimates the quality of verifiers by adapting a method-of-moments (MoM) approach [1], and then optimizes the decision rule (e.g., logistic regression). Then, based on the optimized decision rule, the final selection is made. The authors demonstrate the effectiveness of FUSE following the previous test-time scaling work [2] or using  the recent challenging reasoning benchmarks such as Humanity's Last Exam and IMO Shortlist. With these extensive evaluations, FUSE outperforms unsupervised selection baseline and yields matched performance with supervised baselines.

[1] Jaffe et al., Estimating the Accuracies of Multiple Classifiers Without Labeled Data, AISTATS 2015
[2] Saad-Falcon et al., Weaver: Shrinking the Generation-Verification Gap by Scaling Compute for Verification., NeurIPS 2025

**Compliance With Llm Reviewing Policy:**

Affirmed.

**Final Justification:**

My concerns are sufficiently addressed, and therefore I maintain my original rating toward the acceptance.

**Key Questions For Authors:**

Please address the comments in weaknesses.

**Limitations:**

No limitations were discussed in the draft. The authors should add them in the revised draft.

**Strengths And Weaknesses:**

*Strengths*
- **Clear presentation**: Overall, the draft is well-written.
- **Novel and effective method for important problem**: Combining multiple verifiers to select the most promising output is an important direction for better test-time scaling, which is a timely and important problem. While the core idea of proposed method is inspired by [1], the authors validate that the proposed modifications are key for the empirical gains. Also, across three different setups, the proposed FUSE continuously exhibits the empirical effectiveness.

*Weaknesses*
- **Computational cost**: In FUSE, it seems that there are many parts requiring large computations. For example, finding $\tau^{\*}$ (Line 2 in Algorithm 1) requires coordinate descent. Also, applying theorem 2.3 to $\tilde{V}$ (Line 4 in Algorithm 1) involves the computation over $m \times m \times m$ tensor, which could be costly depending on $m$. Finding $\theta^{\*}$ with logistic regression might incur additional computations. However, there are no discussion regarding such computational costs. It should be presented to clarify the empirical usefulness of FUSE.
- **Reduced gains with more challenging setups**: While the gain from FUSE is significant in the setups from [1], it is reduced in Humanity’s Last Exam and IMO Shortlist. For example, in IMO Shortlist, FUSE shows the identical performance with naive ensemble despite the complexity and additional computations. While the authors discuss such limited gain might be from the inefficiency of verifiers in E.5, it raises the concerns of this framework on challenging scenario where there are no strong verifiers.
- **Advantages and robustness with more verifiers**: All the experiments are conducted with fixed number of candidate responses and number of verifiers. However, to further improve the performance, it's natural to consider to enlarge the number of verifiers, which can be better or worsen than existing verifiers. In this context, it would be interesting if the authors can show that (1) the proposed method is more effective with large number of strong verifiers and (2) more robust even though many weak verifiers are included.

*Minor comments*
- **Room for improving the presentations**: For example, Figure 1 is mentioned in p1, but appeared in p3. Also, section title is not capitalized. Duplicated citations with same paper [2].
- **No oracle**: No pass@K (oracle) are presented in Table 3 and Table 4. It would be better to include this similar to Tables 1 and 2.

[1] Jaffe et al., Estimating the Accuracies of Multiple Classifiers Without Labeled Data, AISTATS 2015
[2] Saad-Falcon et al., Weaver: Shrinking the Generation-Verification Gap by Scaling Compute for Verification., NeurIPS 2025

---

> ### Author Rebuttal · Authors · 2026-03-31
>
> We thank the reviewer for their thoughtful review and are glad that they found the draft well-written, regard FUSE as a “novel and effective method” for an important problem, and find that the results demonstrate “empirical effectiveness.”
>
> We address the reviewer’s remaining concerns below:
>
> > W1 (Computational cost): "In FUSE, it seems that there are many parts requiring large computations...It should be presented to clarify the empirical usefulness of FUSE."
>
> We expect generation of verifier scores to essentially always be the dominant computational cost in running FUSE. That is, the main cost is not the method itself, but the acquisition of the data it operates on. This is because modern "verifiers" can be very large; they are often themselves language models.
>
> (Whether using multiple verifiers is practical is ultimately context-dependent, but in settings where one is particularly sensitive to accuracy of verification, improving the quality of the verifier ensemble can be viewed as a potential scaling axis. That is, one hopes to invest compute for increased performance; methods like FUSE help to reliably realize such gains.)
>
> At a lower level (ignoring the dominant cost of data acquisition) -- we found the order-three tensor manipulations and logistic optimization (via gradient descent) to be essentially instantaneous on any modern, non-specialized hardware (e.g. laptop CPUs). Much of this is attributable to the fact that while e.g. the tensor operations are $O(m^3)$, in any realistic setting, including those in our paper, $m$ is in the low double digits at most. Coordinate descent can be naively non-trivial with such values of $m$, but can be sped up by exploiting tricks specific to binarization (e.g. one only needs to consider parameter values in the empirical support of each verifier). It is further worth noting that if one picks a continuous transformation family (e.g. sigmoid), gradient descent can be used in place of coordinate descent.
>
> We will add this discussion to the paper.
>
> > W2 (Reduced gains with more challenging setups): "While the gain from FUSE is significant in the setups from [1]...it raises the concerns of this framework on challenging scenario where there are no strong verifiers."
>
> This is a good point. Broadly, we do not expect (and certainly do not claim) that FUSE can extract signal when no such signal exists. When all verifiers are poor, there is simply not much that can be done. In the extreme case of most verifiers being worse than random, the problem is even unidentified from a statistical perspective; see our inclusion and discussion of Assumption A.1 from Jaffe et al. (2015).
>
> With that said, both HLE and IMO Shortlist exhibit some special structure which we believe to have explanatory power for reduced gains. In the IMO Shortlist setting, the culprit for the naive $\approx$ FUSE equivalence is not the inefficiency of verifiers per se, but that when verifiers are homogeneous in strength and conditionally independent, the naive ensemble is near-optimal*. Indeed, our verifiers in that setting exhibit homogeneous balanced accuracies and are nearly conditionally uncorrelated (see Figures 6/7 in Appendix). The fact that FUSE matches naive ensemble can therefore be viewed as a sign of robustness (as opposed to, e.g., failure to capture signal).
>
> (* Under conditional independence, the optimal ensemble depends, to first order, only on balanced accuracies; see Parisi et al., Eq. (14) and surrounding discussion. When balanced accuracies are equal, this reduces to the naive ensemble.)
>
> Finally, for HLE, we view the scope for gains as somewhat limited by the extreme bimodality of the (# correct responses out of 50) distribution on the benchmark. For instance, 27.89% of questions have exactly 1 or 50 correct responses, so on over a quarter of the dataset, it is either trivial or maximally difficult to identify a correct response.
>
> > W3 (Advantages and robustness with more verifiers): "All the experiments are conducted with fixed number of candidate responses and number of verifiers...even though many weak verifiers are included"
>
> We will take this into consideration in the final draft. It is difficult for us to execute additional experiments that involve adding verifiers in the rebuttal time-frame (as data generation requires either API spend or significant GPU inference).
>
> > W4 (Room for improving the presentations):
>
> We thank the reviewer for pointing out these presentation issues—we will fix them all in the final version of the paper.
>
> > W5 (No oracle):
>
> We will update Tables 3 and 4 to include oracle pass@K values.
>
> > “No limitations were discussed in the draft. The authors should add them in the revised draft.”
>
> Noted – we will do so. In particular, we will discuss the computation-heavy nature of obtaining repeated generations and verifications and emphasize that FUSE cannot salvage settings in which verifiers provide no signal.

---

> > ### Author Rebuttal · Reviewer_qT59 · 2026-04-03
> >
> > I appreciate the authors for the detailed response. My concerns are sufficiently addressed, and therefore I maintain my original rating toward the acceptance.

---

### Decision · Program_Chairs · 2026-04-30

**Decision:**

Accept (regular)

**Comment:**

All reviewers agree the paper is technically solid with strong empirical results and a well-motivated approach to unsupervised verifier ensembling. Concerns about computational cost, complexity, and diminishing gains in harder settings are valid but adequately addressed in the rebuttal and do not undermine the core contribution. Overall, this is a solid and timely contribution that merits acceptance.